# Heat guiding and focusing using ballistic phonon transport in phononic nanostructures

Roman Anufriev[1,*], Aymeric Ramiere[1,2,*], Jeremie Maire[1] & Masahiro Nomura[1,3]

Unlike classical heat diffusion at macroscale, nanoscale heat conduction can occur without energy dissipation because phonons can ballistically travel in straight lines for hundreds of nanometres. Nevertheless, despite recent experimental evidence of such ballistic phonon transport, control over its directionality, and thus its practical use, remains a challenge, as the directions of individual phonons are chaotic. Here, we show a method to control the directionality of ballistic phonon transport using silicon membranes with arrays of holes. First, we demonstrate that the arrays of holes form fluxes of phonons oriented in the same direction. Next, we use these nanostructures as directional sources of ballistic phonons and couple the emitted phonons into nanowires. Finally, we introduce thermal lens nanostructures, in which the emitted phonons converge at the focal point, thus focusing heat into a spot of a few hundred nanometres. These results motivate the concept of ray-like heat manipulations at the nanoscale.

[1] Institute of Industrial Science, the University of Tokyo, Tokyo 153–8505, Japan. [2] Laboratory for Integrated Micro Mechatronic Systems/National Center for Scientific Research-Institute of Industrial Science (LIMMS/CNRS-IIS), the University of Tokyo, Tokyo 153–8505, Japan. [3] PRESTO, Japan Science and Technology Agency, Saitama 332–0012, Japan. * These authors contributed equally to this work. Correspondence and requests for materials should be addressed to R.A. (email: r.l.anufriev@gmail.com) or to M.N. (email: nomura@iis.u-tokyo.ac.jp).

Studies of nanoscale heat conduction in semiconductors are largely motivated by overall miniaturization of microelectronic devices[1] and by the search for particular nanostructures suitable for thermoelectrics[2], heat localization[3], thermal rectification[1,4] and other applications[5]. However, at the nanoscale, heat conduction cannot be described by classical diffusion along temperature gradients because individual phonons can travel in straight lines without energy dissipation for hundreds of nanometres. Such point-to-point propagation of phonons between diffuse scattering events is called ballistic phonon transport[6–8] and has recently been detected[9–14] in various nanostructures.

Yet, practical application of the ballistic phonon transport remains challenging as different phonons travel in different directions. Once control over directionality is achieved, the possibility to guide and locally apply heat without dissipation can be used in biomedicine[15–18], thermoelectrics[19], phase-change material technology[20,21], chemical reaction control[22] and virtually wherever wireless nano-heaters are required.

A solution may be found in the novel class of thermal materials called phononic crystals[23–25], nanostructures that typically consist of thin films with two-dimensional arrays of holes, which can be designed to control the propagation of phonons. Although room-temperature heat conduction in phononic crystals is mostly dominated by diffuse surface scattering of phonons on the holes[26–28], some phonons can still travel ballistically over hundreds of nanometres, for example, in the cross-plane direction[9].

In this work, we aim to control ballistic heat conduction in the in-plane direction by shaping the paths of ballistic phonons via nano-patterning. Using micro time-domain thermoreflectance (µ-TDTR) experiments and Monte Carlo simulations, we show that silicon phononic crystals can orient the propagation of ballistic phonons in one direction and that this effect can be used for heat guiding and focusing at the scale of a few hundred nanometres.

## Results

**Experimental technique.** To study nanoscale heat conduction, we used an originally developed µ-TDTR technique[29,30] (Methods). This all-optical experimental method is a powerful tool for contactless thermal measurements on a large number of samples. Figure 1 shows the schematics of our µ-TDTR set-up and a typical sample. The samples are mounted in a high-vacuum helium-flow cryostat that enables a precise temperature control. A microscope objective lens focuses two laser beams on an aluminium pad placed on the top of each sample; the pulsed pump beam periodically heats the aluminium pad, while the change in its reflectance ($\Delta R/R$), caused by the heating, is monitored by the continuous-wave probe beam. Since the change in the reflectance is proportional to the change in the temperature via the thermoreflectance coefficient, we can record relative changes in the temperature ($\Delta T/T$) of the aluminium pad in time ($t$). Figure 1b shows that a measured thermal decay curve can always be well fitted by an exponential decay $exp\,(-t/\tau)$; thus, the only quantity that characterizes each sample is the thermal decay time ($\tau$), the time for heat to dissipate from the aluminium pad through the sample. To ensure dissipation only through the structure of interest, the investigated structures are suspended, as shown in Fig. 1c.

To create such structures, we used a standard top–down approach on a silicon-on-insulator wafer with a 145-nm-thick top layer (Methods). First, we deposited $4 \times 4\,\mu m^2$ aluminium pads in the centres of the future structures. Next, the phononic structures were formed by electron-beam lithography followed by reactive

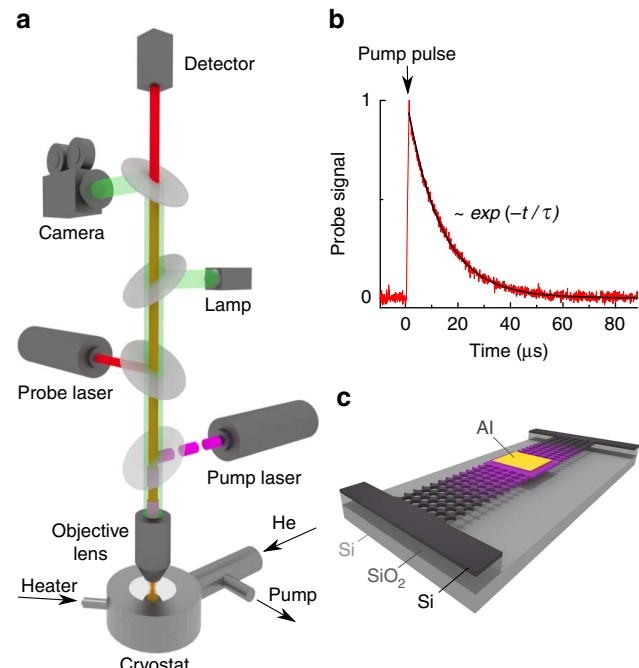

**Figure 1 | Experimental set-up and a typical sample for thermal decay measurements.** (**a**) Pump and probe beams are focused on the sample by an objective lens; a detector monitors changes in the reflection of the probe beam ($\Delta R/R$) in time caused by the pulse of the pump beam. (**b**) The signal is fitted by an exponential function. (**c**) The schematic of a typical air-bridged sample.

ion etching. Finally, the underlying $SiO_2$ layer was removed using vapour hydrofluoric etching, creating 5-µm-wide and 145-nm-thick air-bridged structures, with the length depending on the type of the structure: 25 µm for aligned and staggered samples, 12 µm plus the nanowire length for nanowire-coupled samples and 16 µm for thermal lenses (Supplementary Fig. 1).

**Heat conduction in aligned and staggered lattices.** First, we investigated the possibility of in-plane ballistic heat conduction in phononic crystals. One method to detect ballistic phonon transport is to compare the structures in which phonons can freely travel in straight lines with those in which the direct path of phonons is reduced or even blocked. Thus, we studied two types of hole arrangement: an aligned (square) lattice and a staggered lattice, in which every second row of holes is shifted by a half period from its position in the aligned lattice. The staggered lattice should not be confused with a hexagonal (triangular) lattice[30], in which the density of holes is higher, or with a slightly disordered lattice[31], in which passages between holes remain open. To assess heat conduction at different scales, we fabricated samples with periods of 160, 200, 280, 350 and 500 nm, shown in Fig. 2a–d, and several diameter-to-period ($d/a$) ratios for each period.

Generally, heat dissipation becomes slower as the diameter-to-period ratio is increased or as the period is reduced because the material volume is reduced and the phonon scattering surface area is increased[30]. Here, the samples with the aligned and staggered lattices have the same volume and the same surface area. For this reason, at the microscale ($a = 500$ nm), the thermal decay times ($\tau$) measured on the samples with the aligned and staggered lattices were indistinguishable (Fig. 2e), at least as long as the diameter-to-period ratio was below 0.75. This result is consistent with the pioneering work by Song and Chen[32] on

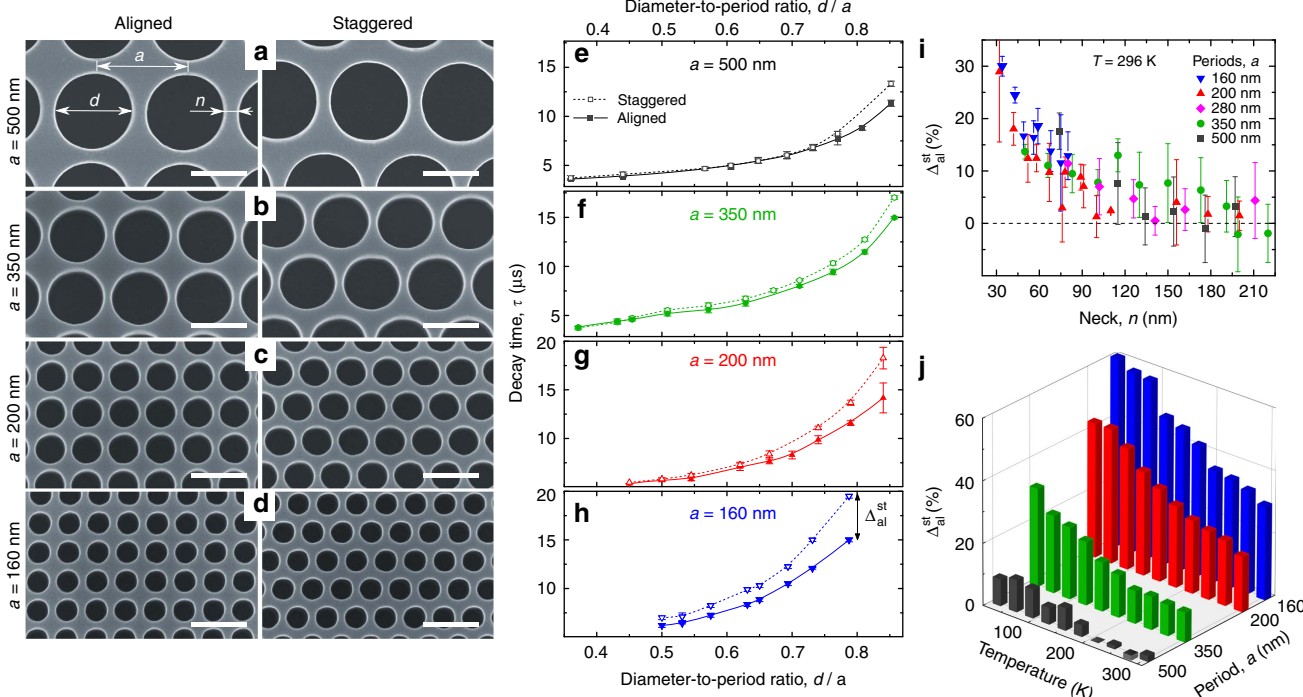

**Figure 2 | Thermal conduction in aligned and staggered lattices.** (**a–d**) The SEM images of typical aligned and staggered lattice samples with periods of 500, 350, 200 and 160 nm. The scale bars, 300 nm. (**e–h**) The decay times measured at room temperature (296 K) for the aligned samples deviate from those for the staggered ones as the period is changed from 500 to 350, 200 and 160 nm and as the diameter-to-period ratio is increased. The lines are guides for the eye. (**i**) The difference between the decay times of the aligned and staggered samples ($\Delta_{Al}^{st} = (\tau_{st} - \tau_{al})/\tau_{al}$) correlates with the neck (*n*), and (**j**) strengthens by a factor of two as the temperature is decreased to 30 K. The error bars depict the s.d. in multiple measurements (Methods) and are less than the size of most data points in (**e–h**) and less than ± 7% in (**j**).

aligned and staggered microporous structures ($a = 4\,\mu m$ and $d/a = 0.57$) and confirms that microscale heat conduction is mostly diffusive[33]. Yet, even in these long-period samples, the decay times in the aligned lattice deviated from those in the staggered lattice for $d/a > 0.75$. This deviation was increasingly pronounced as the period was reduced to 350 nm (Fig. 2f) and further to 200 nm (Fig. 2g). Finally, as we scaled the structures down to 160 nm in period, the difference between the aligned and staggered lattices became evident for all the measured samples (Fig. 2h). In other words, at the nanoscale, heat dissipated faster through the aligned lattice than through the staggered lattice.

The difference between the aligned and staggered lattices did not simply appear at a certain porosity, regardless of the scale, but seemed to depend on some characteristic dimension of the structure. One such dimension, impacting heat transfer in phononic crystals[30,34], is the neck (*n*), the distance between two adjacent holes. Figure 2i shows that the difference between the decay times in the aligned ($\tau_{al}$) and staggered ($\tau_{st}$) samples $\Delta_{al}^{st} = (\tau_{st} - \tau_{al})/\tau_{al}$ seems to correlate with the neck size: regardless of the period, a clear difference appears only when the neck becomes smaller than 100 nm and increases up to 30% as the neck is reduced down to 30 nm. In terms of effective thermal conductivity, the value in the smallest samples ($a = 160$ nm, $d/a = 0.79$) was reduced from $7.28\,W\,m^{-1}\,K^{-1}$ in the aligned lattice to $4.36\,W\,m^{-1}\,K^{-1}$ in the staggered lattice (Supplementary Figs 2–4 and Supplementary Note 1). Thus, staggering is an effective method to reduce the thermal conductivity and can further improve the performance of thermoelectric devices based on phononic nanostructures[35].

Next, we repeated the measurements at different temperatures in the 30–296 K range on four characteristic pairs of samples, one of each period ($d/a \approx 0.73$ for $a = 500$ and 350 nm; $d/a \approx 0.79$ for $a = 200$ and 160 nm). We found that the effect was temperature-dependent and strengthened by a factor of two as the temperature was decreased from 296 to 30 K (Fig. 4j). At 30 K, the difference $\Delta_{al}^{st}$ eventually appeared even in the largest samples, while in the smallest ones it became as high as 60%.

**Ballistic heat conduction in phononic crystals.** These observations cannot be explained in terms of classical heat diffusion (Supplementary Fig. 5); thus, we assume that at nanoscale heat no longer spreads diffusively but propagates partly ballistically. Indeed, in the aligned lattice, phonons could in principle travel ballistically in the passages between the holes, whereas in the staggered lattice, these passages are reduced or even blocked completely.

To verify this hypothesis, we simulated the propagation of phonons in our samples ($a = 350$ nm, $d/a = 0.85$, $\eta = 2$ nm) using a specially developed three-dimensional Monte Carlo technique (Methods). Due to the Debye approximation used in our Monte Carlo algorithm, we conducted all simulations for the temperature of 4 K. We stress that performing these simulations at 30 K would not change the main observations that follow.

Figure 3 shows the distributions of thermal energy in the aligned and staggered structures and the corresponding spatial and angular distributions of phonons after ten rows of holes. In the staggered lattice, phonons are scattered by staggered holes; this scattering reveals itself as faint yellow regions just under the upper row of holes (Fig. 3a). Thus, at the end of the structure, phonons are uniformly scattered in space and have a broad range of exit angles (Fig. 3b).

However, in the aligned lattice, phonons develop a certain directionality, and fluxes of heat appear in the passages between

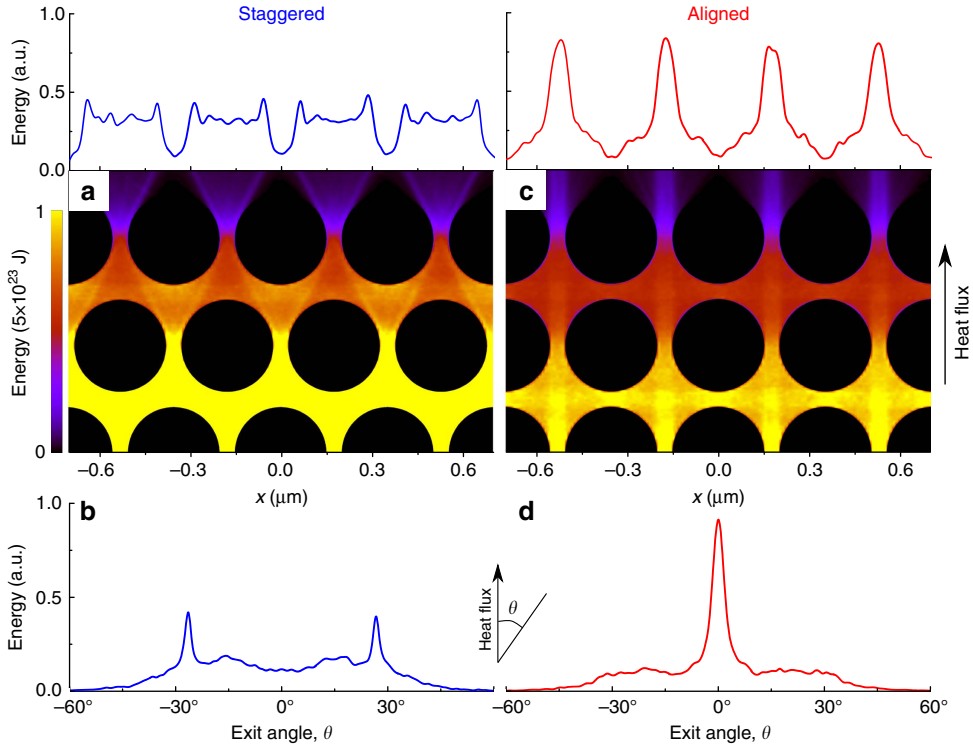

**Figure 3 | Heat flow directionality in aligned and staggered lattices.** Monte Carlo simulations predict rather uniform spatial and angular energy distributions in the staggered lattice (**a,b**), but strong anisotropy in the aligned lattice (**c,d**).

the holes (Fig. 3c). As a consequence, the spatial distribution peaks in front of the passages and the angular distribution sharply peaks at zero degrees (Fig. 3d). In addition, we found that this directionality strengthens as the number of rows of holes is increased and as the neck is reduced (Supplementary Fig. 6 and Supplementary Note 2).

Thus, ballistic phonon transport can indeed occur in phononic crystals with an aligned lattice. Let us consider this process in more details. Ballistic transport suggests that phonons can travel, at least until a phonon–phonon scattering event, experiencing either no scattering at all or only specular surface reflections, as opposed to diffuse scattering in random directions[36]. The probability ($p$) of specular scattering depends[37] on the phonon wavelength ($\lambda$), root mean square surface roughness ($\eta$) and normal incidence angle ($\alpha$) as $p = exp \ (-16 \ \pi^2 \ \eta^2 \ cos^2\alpha/\lambda^2)$. The range of the phonon wavelengths depends on the temperature and shortens from 10–100 nm at 4 K to 0.5–6 nm at room temperature[38–40]. Assuming, for instance, a dominant wavelength of $\lambda = 2$ nm and $\eta = 2$ nm (Methods), we can roughly estimate that even at room temperature the scattering on hole surfaces can be partly specular ($p > 0$) if phonons approach the surfaces tangentially ($\alpha > 80°$). Such tangential incidences often occur[41,42] in the passages between the holes of the aligned lattice as phonons develop directionality passing through the structure (Supplementary Fig. 6).

Indeed, even at room temperature, similar heat fluxes between the holes of phononic crystals seem to occur in simulations based on the Boltzmann transport equation for phonons[42–44]. Tang et al.[42] showed that the impact of specular surface reflections grows as phononic crystals are scaled down. Assuming purely specular surface reflections ($p = 1$), Tang et al.[42] predicted a 15% difference between the thermal conductivities of aligned and staggered structures, whereas assuming purely diffuse scattering ($p = 0$) other theoretical

works[41,45] captured no significant difference. Recently, Hao et al.[46] considered aligned lattices in both purely diffusive and purely specular scattering approximations; they found that the difference between the thermal conductivities in these two cases increases as the neck becomes smaller and at the neck of 30 nm, the difference reaches 32%, which is consistent with our results. Thus, our experimental data can be explained only by the presence of specular surface reflections and ballistic phonon transport in the passages between the holes.

To explain the observed temperature dependence of this effect, we should take into account that the phonon wavelengths become longer at lower temperatures, and thus the range of the incident angles, at which phonons can reflect specularly, becomes wider (Supplementary Fig. 7 and Supplementary Note 3). Hence, the number of ballistic phonons is larger at low temperatures. Likewise, the bulk mean free path, the average distance that phonons can travel until a phonon–phonon scattering event, lengthens at lower temperatures[47–49], and thus, phonons can travel ballistically over longer distances.

Additionally, some long-wavelength phonons can be reflected specularly at all incidence angles (Supplementary Fig. 7) and thus experience interference due to the periodicity of holes[28,50]. Although at room temperature, such phonons represent only a small part of the phonon spectrum in our system; at very low temperatures, their portion may become significant. At 4 K, for example, such phonon interference can suppress the thermal conductivity by 10–20% (ref. 31). However, this phenomenon appears in aligned and staggered lattices alike because both are strictly periodic, and thus, it should not affect the observed difference between the lattices (Supplementary Fig. 8).

**Coupling ballistic phonons into nanowires.** The observed directionality of phonons in the aligned lattice implies that such phononic nanostructures can act as guides and directional

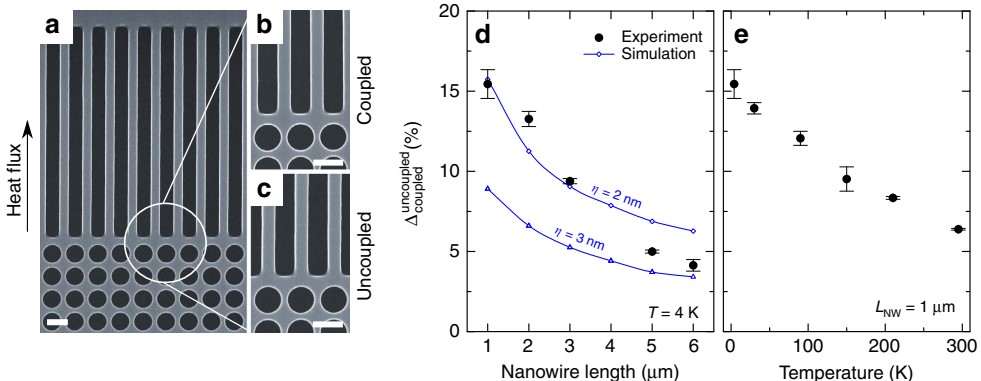

**Figure 4 | Phonon coupling into nanowires. (a)** SEM image of a typical nanowire-coupled phononic crystal sample. Close-up views showing the difference between the coupled (**b**) and uncoupled (**c**) samples. The scale bar, 300 nm. The experimentally measured difference between the nanowire-coupled and uncoupled samples ($\Delta_{\text{coupled}}^{\text{uncoupled}} = (\tau_{\text{uncoupled}} - \tau_{\text{coupled}})/\tau_{\text{coupled}}$) is (**d**) length- and (**e**) temperature-dependent, which shows that the phononic crystal acts as a directional source of ballistic phonons. The error bars show the s.d. between different measurements on the same sample. The Monte Carlo simulations, represented by the lines, predict similar trends and show that the relative difference is roughness-dependent.

sources of ballistic phonons. To demonstrate the directionality of phonons exiting from the phononic crystals, we fabricated our second set of samples, in which phononic crystals with ten rows of holes in the aligned lattice ($a = 320$ nm, $d/a \approx 0.84$) were connected to nanowires with a width of 120 nm and length in the 1–6 µm range, as shown in Fig. 4a. These samples were of two types: coupled and uncoupled. In the coupled samples, the nanowires were placed in front of the passages between the holes (Fig. 4b), in the maxima of the phonon spatial distribution discussed above. In this configuration, phonons with small exit angles could ballistically pass from the phononic crystal directly into the nanowires. In the uncoupled samples, the nanowires were placed behind the holes (Fig. 4c); thus, the direct passages into the nanowires were blocked.

The type of heat conduction in nanowires is known to be length-dependent[12,13,51], being mostly ballistic in silicon nanowires of a few micrometres in length at low temperatures[52], but becoming diffusive as the nanowire length is increased. This fact makes our experiment more challenging because we can expect not only a significant difference between the decay times of the coupled and uncoupled samples, but also a dependence on the nanowire length.

Figure 4d shows the relative difference $\Delta_{\text{coupled}}^{\text{uncoupled}} = (\tau_{\text{uncoupled}} - \tau_{\text{coupled}})/\tau_{\text{coupled}}$ between the decay times measured on the nanowire-coupled ($\tau_{\text{coupled}}$) and uncoupled ($\tau_{\text{uncoupled}}$) samples as a function of the nanowire length ($L_{\text{NW}}$) at 4 K. In the samples with short nanowires ($L_{\text{NW}} = 1$ µm), heat dissipated 15% faster through the coupled structure than through the uncoupled one. However, as the nanowires lengthened, this difference gradually decreased and almost disappeared at a length of 6 µm. This experiment shows that phononic crystals indeed create fluxes of phonons parallel to the nanowire axis. These fluxes enforce ballisticity in short nanowires and thus, cause the difference between coupled and uncoupled samples, though the effect is partly masked by the phononic crystal part before the nanowires. This difference progressively disappears as the heat conduction eventually becomes diffusive in long nanowires[12,51], regardless of the initial directionality.

To show that faster heat dissipation in the coupled configuration comes from the directionality of phonons entering the nanowires, we simulated phonon transport in the nanowires (Methods) with two distributions of the angles, at which phonons enter the nanowires: the distribution shown in Fig. 3d, representing the coupled configuration, and uniform distribution,

representing the uncoupled one. By measuring the average time that phonons stay in the nanowires in both cases, we can calculate $\Delta_{\text{coupled}}^{\text{uncoupled}}$ and compare the simulation and experimental results. Although the simulation results (Fig. 4d) depend on the surface roughness, the trends are qualitatively consistent with our experimental data.

Moreover, the effect weakens as temperature is increased (Fig. 5c), which shows again that the effect is linked to the phonon wavelength and mean free path. Indeed, since the bulk mean free path in silicon at room temperature is in the 0.1–10 µm range[38,47,49,53], less than half of the phonons can traverse ballistically even 1-µm-long nanowires[38].

**Heat focusing.** Hence, the experiments described above showed that phononic crystals can act as a directional source of ballistic phonons and therefore can be used for a variety of applications, one of which is heat localization. To show its practical realization, we propose a converging thermal lens consisting of circular rows of holes with various diameters, but identical necks, as shown in Fig. 5a. In such a structure, our Monte Carlo simulations predict that the heat fluxes from the passages between the holes will converge at the focal point, located 0.5 µm away from the structure, forming a hot spot with a full width at half maximum of 115 nm (Fig. 5b). To demonstrate this phenomenon, we fabricated a set of samples with the same converging lens but different positions ($\delta$) of a narrow slit (Fig. 5a), through which heat can dissipate from the system. Additionally, we fabricated two sets of reference samples: the first with inverted (diverging) lenses, shown in Fig. 5c, and the second without any holes.

In the reference samples, heat is evenly dispersed in all directions, as shown by the simulation in Fig. 5d. Consequently, for both types of reference samples the measured thermal decay times are nearly independent of the slit position (Fig. 5e–f). However, in the converging lenses, heat dissipates faster when the slit is at the focal point ($\delta = 0$ µm), but the dissipation slows down as the slit is moved aside (Fig. 5g). These results cannot be explained in terms of purely diffusive transport regime (Supplementary Fig. 9) and thus indicate that the converging thermal lens indeed focuses heat in the focal point: the closer the slit to the focal point, the faster heat can escape. To compare the experimental data with predictions of Monte Carlo simulations (Methods), in Fig. 5e–g we plot the simulated decay

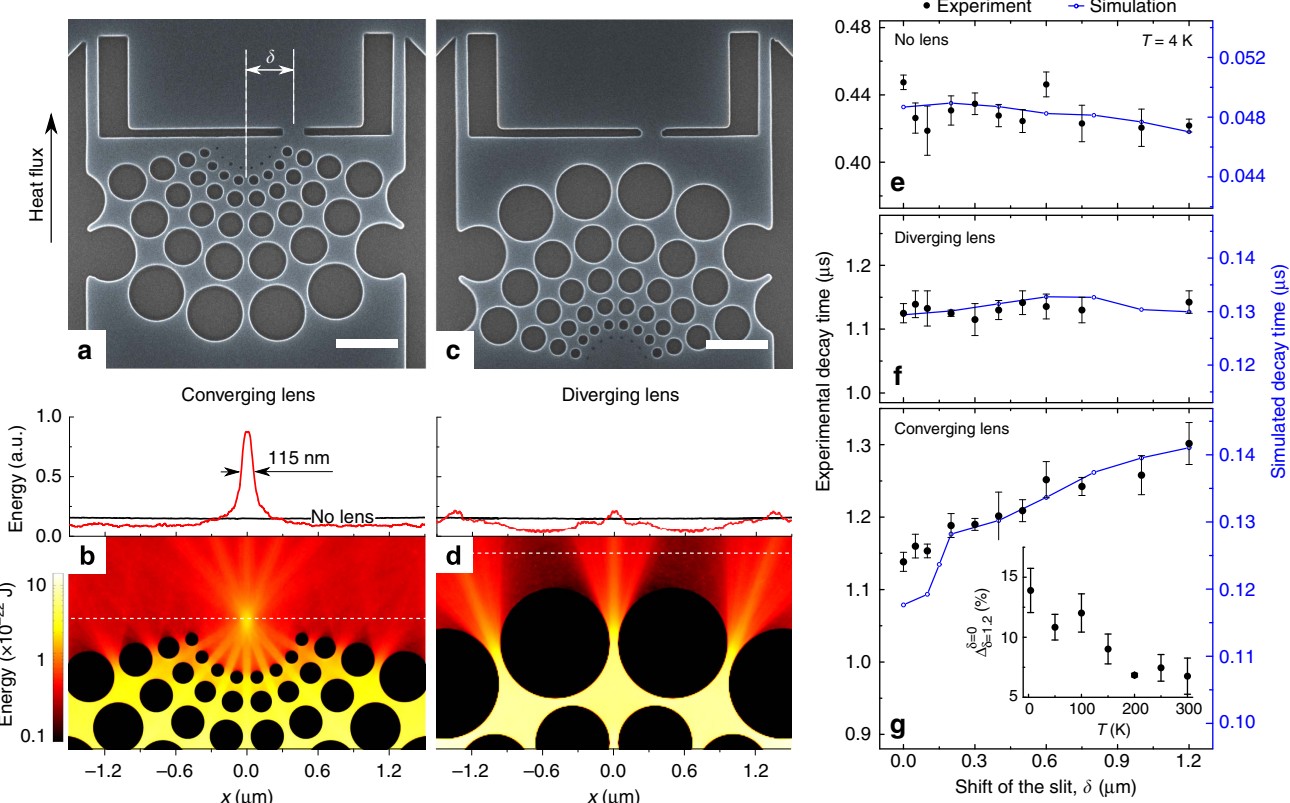

**Figure 5 | Heat focusing.** SEM images of (**a**) converging and (**c**) diverging thermal lens samples with slits for heat dissipation. The scale bar, 1 μm. The Monte Carlo simulations predict (**b**) the formation of a hot spot in the focal point of the converging lens, but (**d**) dispersion of heat in the diverging lens. Both, the experimental (left axis) and simulated (right axis) decay times show no impact of the slit position in the reference samples without (**e**) a lens or with the diverging lens (**f**), but a strong dependence in the samples with the converging lens (**g**). The inset shows that the relative difference between the decay times at $\delta = 0$ and 1.2 μm decreases with temperature. The error bars show the s.d. between different measurements on the same sample.

times on the right axis keeping the same relative scale as for experimental decay times. Although in absolute values, the simulated decay times are much shorter than the experimental ones (Supplementary Fig. 10 and Supplementary Note 4), the trends are generally in good agreement. Interestingly, for the converging lens, the simulations show a drop of the decay time when the slit is close to the focal point because phonons can quickly escape through the slit as long as the focal point is within the slit. However, this feature is much less pronounced in the experimental data, which might suggest that the real hot spot is wider than that predicted by the simulations. A more direct observation of the hot spot could be achieved by thermal mapping techniques and should be the subject of a future work.

The strength of the heat focusing effect also depends on the temperature. The inset in Fig. 5g shows that the difference $\Delta_{\delta=1.2}^{\delta=0} = (\tau_{\delta=1.2} - \tau_{\delta=0})/\tau_{\delta=0}$ between the decay times at different slit positions ($\delta = 0$ and 1.2 μm) decreases from 14% at 4 K to ∼6% at room temperature because the phonon directionality weakens with the temperature, as discussed above. In the Monte Carlo simulations, we also observed that the hot spot gradually disappears as phonon wavelengths are increased (Supplementary Fig. 11). However, since the simulations are strictly valid only below 20 K, the temperature-dependent simulations should be the subject of future work.

Another possible application of the thermal lens structure is thermal rectification. Indeed, if the slits are to be placed on both sides of the lens, the heat propagation should depend on the slit position in the direction of the converging lens, but not in the opposite direction. Hence, the structure could act as a tunable thermal diode.

## Discussion

In this study, we showed that ray-like heat manipulations become possible by using phononic nanostructures. First, the comparison of aligned and staggered lattices at different scales and temperatures revealed the possibility of ballistic phonon transport in the aligned lattice. Next, the proof-of-concept experiments showed that these ballistic phonons can even be collected into nanowires, in which the phonons continue to travel ballistically. These results suggest that phononic crystals can guide and emit phonons in a given direction. To show the potential for practical applications, we introduced thermal lens nanostructures and showed the evidence of nanoscale heat focusing. Since such passive local heating does not require any external laser irradiation, it could become an alternative to the conventional photothermal approach[3,54,55]. Moreover, the temperature at the hot spot can be controlled via the design of the lens, and thus, several hot spots with different temperatures can be created on the same wafer. One can imagine various nanostructures that can guide, turn, confine or disperse heat fluxes using different hole patterns. The efficiency of this approach can be improved by using materials with longer phonon mean free path and is limited only by the scale and quality of the structures. Thus, as the advancements in nanofabrication technology enable further scaling down, complete control over directionality of ballistic heat fluxes in nanostructures becomes real in the near future.

## Methods

**Sample fabrication.** All samples were fabricated on a commercially available silicon-on-insulator wafer with a 1-μm-thick buried SiO₂ layer and a 145-nm-thick top monocrystalline (100) silicon layer. First, electron-beam lithography was

**Figure 6 | Surface roughness.** (**a**) AFM image of the top surface shows a surface roughness of less than 0.5 nm. (**b,c**) SEM images of a neck between two holes and the border of a nanowire show that the peak-to-peak surface irregularity is lower than 4 nm bar. (**d**) The cross-sectional SEM image shows vertical hole profiles. The scale bar, 50 nm.

performed (with ZEP 520A as a resist) in order to deposit 125-nm-thick aluminium pads using electron beam physical vapour deposition (Ulvac EX-300), which was followed by the resist lift-off. Next, the structures were formed in the top silicon layer by the second electron-beam lithography followed by reactive ion etching by means of an inductively coupled plasma system (Oxford Instruments Plasmalab System 100 ICP), with $SF_6/O_2$ gas as an etchant. Finally, the buried oxide layer was removed using vapour etching with diluted hydrofluoric acid. The hole diameters were measured via scanning electron microscopy (SEM) for all the samples with an inaccuracy of $\pm 2$ nm.

This study is designed as a comparative between pairs of similar samples; thus, systematic errors are excluded from consideration as most of the geometrical parameters (density of holes, porosity, volume of the material, etc.) are the same in both samples (aligned and staggered, coupled and uncoupled). Moreover, the samples of each set were fabricated simultaneously on the same wafer; thus, we expect no variations in the surface roughness as well as no difference in the rates of internal phonon scattering processes. For the aligned and staggered lattices, two identical sets of samples were fabricated and measured.

**Surface roughness considerations.** Figure 6a shows the atomic force microscope image of the top surface of the sample, obtained using a PRC-DF40P Olympus cantilever with a tip radius of 20 nm. The maximum surface irregularity does not exceed 0.5 nm, and thus, the r.m.s. surface roughness ($\eta$) is significantly below this value and the roughness of the top and bottom surfaces is negligible. Figure 6b,c show the SEM images of a neck between two holes and the side wall of a nanowire (top view), respectively. Although it is impossible to measure the surface roughness by SEM accurately, we estimated that the peak-to-peak surface irregularity does not exceed 4 nm. Since the r.m.s. value of the roughness is less than its maximum amplitude, we assume $\eta \leq 2$ nm. The cross-sectional SEM image (Fig. 6d) also indicates a low surface roughness and vertical hole profiles. See Supplementary Note 3 for the discussion on the impact of the surface roughness and its correlation length on the phonon surface scattering.

**Micro time-domain thermoreflectance.** The samples were placed in a He-flow cryostat mounted on motorized linear stages. This enabled us to perform the measurements in vacuum ($<10^{-4}$ Pa) with precise control over the temperature ($\pm 0.02$ K) and the position of the samples. A red light diode illuminated a large area on the surface of the wafer, making it possible to see the samples. Pulsed pump laser ($\lambda = 642$ nm, length of the pulse is 1 μs, repetition rate is 1 kHz) and continuous-wave probe laser ($\lambda = 785$ nm) beams were focused on an aluminium pad on the top of the studied sample via an optical ($\times 40$) objective. The reflection of the probe beam was continuously monitored by a silicon photodiode detector connected to a digital oscilloscope. Each 1-μs-long pulse of the pump beam caused a leap of the temperature in the aluminium pad ($\sim 300$ nW is absorbed) followed by relaxation as heat dissipated through the sample (the average temperature increase in the structure does not exceed 5 K). This process was detected as a leap in the probe beam signal intensity caused by the change in reflectivity ($\Delta R$), due to the heating, followed by a gradual return to the background value as the heat dissipated. A typical normalized $\Delta R/R$ signal as a function of time is shown in Fig. 1b. This function can be[56] approximated well by an exponential decay. The system recorded and averaged the $\Delta R/R$ decay functions over $10^4$ pulses of the pump beam and automatically fitted the result by an exponential decay function $exp(-t/\tau)$, with $t$ as the time and $\tau$ as a fitting parameter. To ensure the accuracy of the measurement, this iteration was repeated until the s.d. of $\tau$ in the last 30 iterations was less than 1%, at which moment the value of $\tau$ was recorded.

To eliminate inaccuracies that might be caused by the laser beam alignment, each sample was measured twice on different days, whereas converging lens and reference samples were measured three and four times, respectively. Only some of the 160-nm-period samples could not have been successfully measured more than once due to fragility of the structure. Thus, each data point represents an average of several measurements. The error bars are calculated as the s.d., and their values do not exceed $\pm 5$% for most of the samples.

**Monte Carlo simulations.** The algorithm[36] traces trajectories of particle-like phonons (wave packets) in samples identical to those measured experimentally. Phonons are generated on the top of a membrane with random coordinates within

a $4 \times 4$ μm$^2$ region at the centre of the system, representing the aluminium pad, and start moving in random initial directions (Supplementary Fig. 12 and Supplementary Note 5); the angular frequencies ($\omega$) are randomly assigned according to the Planck distribution, calculated within the Debye approximation. When phonons encounter membrane or hole boundaries, the type of surface scattering (specular or diffuse) is determined by the specularity parameter ($p$), as discussed in the main text[37]. We assumed $\eta = 0.3$ nm for the top and bottom boundaries and $\eta = 2$ nm for the hole surfaces, unless stated otherwise. To take into account internal scattering processes, at every moment, each phonon can be scattered in a random direction with a probability of $1 - exp(t/\tau)$, where $t$ is the time since the previous internal scattering event of this phonon and $\tau$ is the characteristic internal scattering time given[57] by $\tau^{-1} = \tau_{impurity}^{-1} + \tau_{normal}^{-1}$, with $\tau_{impurity}^{-1} = 2.95 \times 10^{-45} \omega^4$ and $\tau_{normal}^{-1} = (2\tau_{TA}^{-1} + \tau_{LA}^{-1})/3$, where $\tau_{TA}^{-1} = 9.3 \times 10^{-13} \omega T^4$ and $\tau_{LA}^{-1} = 2.0 \times 10^{-24} \omega^2 T^3$. The simulation time is set so as to reach the steady state when the input power is equal to the power exiting the system. To ensure good statistics, the number of phonons in the system is kept to several millions.

As it is difficult to build Monte Carlo models for too complex geometries, the simulation of the nanowire-coupled samples was conducted in two steps: first, we simulated only phononic crystal part with ten rows of holes and obtained the distribution of the phonon exit angles after these ten rows (Fig. 3); next, we simulated only a single wire, where phonons at the entrance have the distribution of the exit angles obtained on the previous step: phonons in the wires attached in the coupled configuration have a the distriution of the initial angles shown in Fig. 3b, whereas phonons in the wires attached in the uncoupled configuration have a uniform distribution.

The time-dependent simulations are conducted as follows: first, phonons are constantly added to the metal pad area and the energy in the sysem increases until the steady state is reached; then, we stop adding phonons and the energy exponentially decreases. By fitting this decrease, we obtain the decay time. More details can be found in the Supplementary Note 4.

The maps of the energy distribution are obtained as follows: once the steady state is reached, the algorithm starts recording the coordinates and energies $\hbar\omega$ of phonons; the energy is integrated over $10^4$ time steps (10 ns) being recorded at every time step for each phonon into the pixel corresponding to the in-plane coordinate of the phonon. An example of the energy map and energy profiles of a full structure can be found in Supplementary Fig. 13.

**Data availability.** The data and the codes used in this study are available from the corresponding authors upon request.

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

## Acknowledgements

This work was supported by the Project for Developing Innovation Systems of the MEXT, Japan, Kakenhi (15H05869 and 15K13270), PRESTO JST (JPMJPR15R4), Maeda Corporation and Posdoctoral Fellowship program of Japan Society for the Promotion of Science. We also acknowledge Anthony George for assistance with the AFM measurements and Ryoto Yanagisawa for assistance with the sample fabrication.

## Author contributions

R.A. designed the study, fabricated the samples, conducted the measurements, processed and interpreted the experimental data, and wrote the article. A.R. developed the Monte Carlo model, performed the numerical simulations, and contributed to designing the samples. J.M. built the experimental set-up, contributed to the sample fabrication and experimental data processing. M.N. contributed to the design of the study and development of the experimental set-up and supervised the work.

## Additional information

**Competing interests:** The authors declare no competing financial interests.

