## [Peer Review File · Nature Communications]

Reviewers' comments:

Reviewer #1 (Remarks to the Author):

In the submitted manuscript titled, "Heat guiding and focusing using ballistic phonon transport in phononic nanostructures" the authors have attempted to establish the ballistic nature of phonon transport using TDTR experiments at 4K and room temperature in silicon phononic crystals by changing the orientation of the air holes in these nanostructures. Second, the authors attempt to use these phononic crystals as a 'source' of ballistic phonons for transport in nanowires. Lastly, they present a possible application of their findings in form of heat lens with possible rectifying nature. Primary conclusion of the study is that flux can be spatially directed using ballistic phonons that travel in straight lines in phononic crystals of Si. This ballistic nature is enhanced at lower temperatures and for films with smaller necks and can be seen even at room temperature. The authors build on these findings to present these phononic crystals as ballistic phonon sources with potential real-world applications. The study presents an advance in the field of nanoscale heat transport, particularly in the context of phononic crystals. The authors provide evidence to a much discussed question of nature of heat in phononic crystals for a range of temperatures. Additionally, the study includes a partial discussion based on critical parameters like nanoscale surface roughness, phonon wavelengths etc. which provides a concrete basis to draw conclusions from this work.

However, there is one major concern pertaining to the data presented in this manuscript. This concern arises when the data presented in this paper is compared to another manuscript (unpublished) by the authors ["Heat conduction tuning using the wave nature of phonons", accessed from: <https://arxiv.org/abs/1508.04574>]. In the current manuscript it is shown that for a temperature range of 4K to room temperature, the disordered/staggered crystals exhibit a higher decay time (i.e. lower decay rate). This finding lies at the foundation of the present work. On the other hand in the manuscript referenced above, the findings are contradictory and decay rates for disordered/staggered crystals are larger at 4K-12K. In that manuscript, this observation is central to support the conclusions of wave nature of thermal transport. This stark mismatch in observed decay rates for seemingly similar nanostructures should be clarified.

In addition, below is a list of additional points that should be addressed:

1. In the manuscript the modelling results are presented for 4K while one of the important results from the paper is the experimental evidence of existence of ballistic modes at room temperature in phononic crystals. For this reason I would recommend that the modeling results at 300K be included in the manuscript, and the discussion on differences in simulation findings at the two temperatures be included in the text.
2. On the account that the interfaces possess roughness in the samples, how valid/accurate is the use of TDTR technique? What possible variations or errors in measurements owing to diffuse scattering of beams be present in the experimental readings of decay times? What is the thermal conductivity of these nanostructures on the basis of these measurements of decay times? The trends of thermal conductivity should be contrasted against existing works.
3. On Page 13, the possibility of thermal rectification is presented at the end of the main text of the manuscript. Since it is an important and interesting application, I would recommend that, on the very least, a cursory analysis on the degree of rectification in the suggested devices be presented to ascertain the future viability of these thermal lenses for the suggested purpose. Similarly, in conclusions and outlook, the authors discuss a contrast with conventional approach for hot spot creation without quantification and/or reasoning as to why their suggested method would be better.
4. In the second part of study (particularly on Page 10), the impact of surface features of the

nanowire is expected to play a role in thermal transport and the effects of existing roughness in the samples should be included in the discussion. Additionally, even though the focus of the work is not explicitly on surfaces, the authors should be careful that the expression for specularity used for analysis is limited in its scope and is valid only for absence of correlation. For the sake of completeness on the details about specularity particularly in context of thermal transport, I would recommend referencing studies where the surface correlations are included.

5. The authors in the caption to figure 6 state that roughness is less than 5nm while in the text they state that roughness is 4nm. Additionally throughout the main text, it has been reported that $\eta \leq 2\text{nm}$ and the same value is considered for analysis. This should be clarified to avoid confusion.

6. Figure 2 caption contains references to plots f-d (it should say e-h?). Are the error bars smaller than data points wherever they are not plotted in Figure 2 including e-h?

7. At some places the choice of language should be reconsidered to avoid confusion. For example in the second paragraph it seems what the authors are suggesting is that the sole reason that the directionality of flux exists is the diffusive nature of phonons (irrespective of the direction of thermal gradient, entropy change etc). Another term that is used is 'directional fluxes'. Since this term is not standard in the field. it should be explained/defined before being used in the text.

I recommend that the authors submit a revised manuscript addressing all the above concerns.

Reviewer #2 (Remarks to the Author):

This manuscript deals with phonon transport in the ballistic regime. There are two key ideas: (i) the first one is to show that in the ballistic regime it is possible to shape the directionality of heat conduction with additional degrees of freedom in comparison to usual macroscopic diffusion, because there is no requirement of local thermal equilibrium and therefore no local isotropy, (ii) The second idea is that a hot spot can be generated passively by means of a "thermal lens". This is in contrast to the usual knowledge stating that heat flows from hot to cold areas, and that hot spots are consequently required to be located only at the heat sources' locations.

To demonstrate these ideas, three types of structures based on suspended membranes of 145 nm thickness have been fabricated and characterized by optical means in the transient regime: (i) 2D phononic crystals with periodically aligned or periodically disaligned holes, (ii) similar crystals connected to an array of nanowires, and (iii) "thermal lenses" connected to slits. These structures have been fabricated with up-to-date nanofabrication means. The experimental results are supported by Monte-Carlo simulations of phonon transport, probably the only way to describe heat transfer in the ballistic regime for some of the complex structures involved. However, the proof of the creation of a hot spot in-between the hot and cold baths by means of the "lens" stays indirect.

The work is very timely and really novel. This group has produced very interesting results in the last years and seems this time to bring the thermal transport community a new step forward. As a result, this manuscript can certainly deserve publication in Nature Communications if the following questions can be answered in a satisfactory manner and if the comments are taken into account.

Questions and comments

- On the experiments and their analysis

1) No wave character of the phonons is taken into account in the analysis, while at 4 K the same group has previously claimed to have observed some effects. How do these results compare with the previous work: is the disorder previously introduced comparable with the disalignment here

considered? A comment is expected in the manuscript.

2) It is stated that the increase in temperature due to the pump is less than 5 K. At $T_0=4$ K, an increase of $\Delta T=5$ K leads to a value of $\Delta T/T_0 > 100\%$, a situation where rectification effects are expected to be very strong. Such effects are in particular possible in the case of the "lens", which is anisotropic. How could such effects, not necessarily due to ballistic transport, be discarded? Note in particular that it would have been more convincing if reference experiments with lenses in both directions without slits would have been shown.

3) In Fig. 4(d), the simulated lines are quite different from the almost linear trend found in the experiment. What could explain this difference? Is there a way to fit the experimental data with an improved model?

4) In the Suppl. Mat. It is stated that "each data points represents an average between several measurements". Was actually the value fixed for a certain type of structure (then which one?), or in the same Figure two points can be average over a different number of measurements?

5) Figs. 2(e-h), 4(d-e) and S3. The error bars should be set everywhere.

6) The determination of the surface roughness over a square of $10 \times 10 \text{ nm}^2$ is a bit limited. Usually larger areas are considered. In addition, what is the curvature radius expected for the AFM tip that was used?

7) Simulations in the diffusive regime for the cases of the phononic crystals were performed but not for the lenses with the slits. This is missing (can be put in the Suppl. Mat.) to be even more convincing.

- On the Monte-Carlo simulations

8) The Monte-Carlo simulations are apparently producing results in a stationary regime (stationary local energy distribution) while the experiments are transient. It is therefore biased. One could in particular imagine that ballistic phonons travel faster than those which are scattered and generate some diffusive part. Such mechanism is not mentioned anywhere. Some comments are expected here.

9) The Monte-Carlo simulations are performed in 2D, while the thermal transport is 3D. In a suspended membrane the effective average mean free path is reduced in comparison to a real 2D material (because of cross-plane motion): how was this taken into account?

10) The roughness effect at the top and bottom surfaces of the suspended membranes is neglected. In reality there is certainly an η value limited to that of the interatomic distance. This may reduce at least partly the ballistics of the phonon transport. Is this taken into account?

11) For the simulations $\lambda = 3 \text{ nm}$ seems large at room temperature. It is probably closer to 1.5 or 2 nm (maybe even lower), and this has an effect on the specular parameters. How do the results change for smaller wavelength?

12) In Fig. 3, only ten rows are simulated, so that the structure mostly acts as a filter (the mean free path is larger than the full size of the structure) and there is no effective diffusion regime (as in nanowires or suspended membranes that are longer). What is actually the average mean free path at this temperature?

13) It is stated on p.11 that to model the transport in the nanowires Lorentzian and uniform (I guess cosine?) distributions of initial angles are considered. Why not accounting for the results of Fig. 3?

14) How are Normal vs Umklapp processes accounted for in the simulations?

- Completing and improving the manuscript

15) All the simulated local energy maps are given with arbitrary units, while it would be very interesting to know the local effective temperatures, in particular the one of the hot spot in comparison to that of the baths. I suggest to add the units in Figs. 3 (a-d) and 5 (b,d).

16) I suggest adding a figure (at least in the Suppl. Mat.) with the value of the specular factor p as a function of temperature for a given value of roughness η : this would help the reader to understand the effect of temperature.

17) It would be good to have also a picture (not a schematic) of the full device (with Al pad and baths).

18) On p.9, it is written that heat transport is ballistic in nanowires over few micrometers in length. Please give more details here (material, temperature considered) as it is for example not the case for silicon nanowires at room temperature.

19) Fig. 4(e): Strange to have an x-axis that starts below 0 K...

20) In the introduction I suppose that the authors refer to "in-plane transport" and not "cross-plane transport" when citing Ref. [9].

21) Some of the prospects in the last section might be softened by changing "can" into "could"...

22) A reference is expected for the correction factor F in the Suppl. Mat.

23) At many place "heat transport" is written. If "phonon" is not mentioned, I suggest to replace "transport" by "conduction" since other means of thermal transport exist (in particular radiation).

- Errors and typos:

Note that there are still some English language mistakes throughout the manuscript.

Among errors and typos, one can find:

(a) Fig. 2: it is (e-h) and not (f-d)

(b) Fig. 2 (2x) and Fig. S3 (2C): It is "diameter-to period" and not "dimeter-to-period".

(c) p.11: "simulations predict" and not "predicts"

Reviewer #3 (Remarks to the Author):

The authors study thermal transport by phonons in a series of nanofabricated silicon thin films. There are three types of structure: (i) periodic pore arrays (i.e., phononic crystals), which either allow or block a direct line of sight for phonon transport, (ii) phononic crystals with nanowires attached to one end, which either allow or block a direct line of sight for phonons to enter the wires, and (iii) holes of different sizes positioned to focus phonons. In their TDTR-based experiment, a laser heats a metal pad that is in lateral contact with the nanofabricated films. A second laser monitors the temperature response of the pad as thermal energy dissipates through the film. A decay time is extracted and used to compare the structures. Complementary Monte Carlo and continuum simulations are used to interpret the data.

This study is highly original, well conceived, and carefully thought out. The authors have considered structures more complicated than previous works. They do a very good job in

explaining their results in a thoughtful, logical manner. While none of the results are highly surprising, the fact that the authors were able to build structures where clear evidence of ballistic transport and focusing are present is an outstanding achievement.

Comments

1. The comparison between structures is based on the decay time extracted from the TDTR experiments. While I am not arguing that this single number does not resolve important differences between the samples, there is a lot going on in these experiments, where there is a full spectrum of phonon modes. Can the authors comment on what other information might be extracted from their experiments (now or in the future) that would be useful in providing more insight? As of now, there is no way to directly compare the experimental measurements to the simulations, which might provide more insight into the underlying phonon physics. One idea is to analytically solve the heat diffusion equation in one dimension in the samples (with an effective thermal conductivity) and see what time scale emerges.
2. Will the decay always be exponential? What governs that behavior?
3. What exactly is the "thermal energy distribution" plotted in Fig. 3a?
4. Page 8 and related discussion on page 9. The authors specify one phonon wavelength in the discussion of the boundary scattering. It would be better to call this quantity the dominant phonon wavelength and acknowledge that (i) there is a range of values at any temperature, and (ii) this range moves to larger wavelengths as temperature decreases due to the Bose-Einstein occupation.
5. In analyzing the nanowire structures, the effect of the phononic crystal is included in τ , which may decrease the measured difference between the coupled and uncoupled cases. Can the authors conceive of a way to better isolate the nanowire effect?
6. I do not understand the wire simulations (page 11). Are only the wires being considered or is the phononic crystal part of the simulation as well? More detail would be useful, maybe in the Methods section.
7. Page 11: "less than half of the phonons can traverse ballistically even in the 1 micron long wires." What is the basis for this statement?
8. Page 16. The phonon model used is basic. Are the phonon lifetime relations valid at all temperatures considered? Given that the comparison between the experiments and simulations is qualitative, I am not particularly concerned.
9. SI page 1. The thermal conductivities of the silicon and aluminum thin films are taken to be 75 and 237 W/m-K. How were these values obtained?
10. SI page 4. The $F(\phi)$ factor is geometry dependent. I suggest that the authors not provide any form of this equation unless they are explicit about what geometry it applies to.
11. SI page 5 "... have escaped from the phononic crystal during the transient regime." In the Methods section, there is discussion about how data was collected from the simulations once steady state was reached. As such, why would the transient part of the simulation be relevant here?

Reviewer #1

In the submitted manuscript titled, “Heat guiding and focusing using ballistic phonon transport in phononic nanostructures” the authors have attempted to establish the ballistic nature of phonon transport using TDTR experiments at 4K and room temperature in silicon phononic crystals by changing the orientation of the air holes in these nanostructures. Second, the authors attempt to use these phononic crystals as a ‘source’ of ballistic phonons for transport in nanowires. Lastly, they present a possible application of their findings in form of heat lens with possible rectifying nature. Primary conclusion of the study is that flux can be spatially directed using ballistic phonons that travel in straight lines in phononic crystals of Si. This ballistic nature is enhanced at lower temperatures and for films with smaller necks and can be seen even at room temperature. The authors build on these findings to present these phononic crystals as ballistic phonon sources with potential real-world applications. The study presents an advance in the field of nanoscale heat transport, particularly in the context of phononic crystals. The authors provide evidence to a much discussed question of nature of heat in phononic crystals for a range of temperatures. Additionally, the study includes a partial discussion based on critical parameters like nanoscale surface roughness, phonon wavelengths etc. which provides a concrete basis to draw conclusions from this work.

However, there is one major concern pertaining to the data presented in this manuscript. This concern arises when the data presented in this paper is compared to another manuscript (unpublished) by the authors [“Heat conduction tuning using the wave nature of phonons”, accessed from: <https://arxiv.org/abs/1508.04574>]. In the current manuscript it is shown that for a temperature range of 4K to room temperature, the disordered/staggered crystals exhibit a higher decay time (i.e. lower decay rate). This finding lies at the foundation of the present work. On the other hand in the manuscript referenced above, the findings are contradictory and decay rates for disordered/staggered crystals are larger at 4K-12K. In that manuscript, this observation is central to support the conclusions of wave nature of thermal transport. This stark mismatch in observed decay rates for seemingly similar nanostructures should be clarified.

Thank you for your interest in the activity of our group. Indeed, that work considers similar structures and might be relevant to the present study. However, there is no discrepancy.

From the wave point of view, the staggered lattice cannot be regarded as disordered because this lattice is as strictly periodic as the aligned one, so coherence should be similarly present or absent in both lattices; we also calculated phonon dispersions and found that both lattices have equally flattened branches. In addition, the coherence effect is weak (10 – 20%) at 4 K and disappears above 10 K, whereas effects discussed in the present work are stronger and remain present even at room temperature. For these reasons, we think that coherent effects can hardly affect data in the present work, but the effects of this work do affect the results in that unpublished work. Indeed, from the particle point of view, the direct passages between the holes remains open even in the 6% disordered lattice (in which coherence is already gone), whereas in staggered lattice the passages are closed; hence the strong increase in decay time in the staggered lattice, but only slight change in the disordered lattice (in incoherent regime).

This issue is addressed in the Supplementary Information of that unpublished work. The figure below shows that the Monte-Carlo simulations predict a slight increase in decay time when disorder is increased in 2D phononic crystals. This increase in decay time happens exactly because the holes start closing the passages as disorder is increased, so the ballistic transport decays. However, even in strongly disordered lattice, the passages remain rather open.

To clarify this, we have added on pages 4-5: “Note that the staggered lattice should not be confused with a hexagonal (triangular) lattice, in which the density of holes is higher, or with a slightly disordered lattice, in which passages between holes remain open.”

Moreover, we added on page 9: “Additionally, at low temperatures, some low-frequency phonons can be reflected specularly at all incidence angles and thus experience interference due to the periodicity of holes. At 4 K, for example, this effect can suppress thermal conductivity by 10 – 20 %. However, this phenomenon impacts aligned and staggered structures alike, as both are strictly periodic, thus it does not affect the observed difference between the lattices”

In addition, below is a list of additional points that should be addressed:

1. In the manuscript the modelling results are presented for 4K while one of the important results from the paper is the experimental evidence of existence of ballistic modes at room temperature in phononic crystals. For this reason I would recommend that the modeling results at 300K be included in the manuscript, and the discussion on differences in simulation findings at the two temperatures be included in the text.

The problem with room temperature Monte-Carlo simulations is that it is more complex and that the results create more questions than answers. For example, one problem is which phonon spectrum to use at room temperature? On the one hand, we can just calculate the Plank distribution, and most phonons will have wavelengths below 1 nm, which is not very realistic. On the other hand, we can take first principle calculations by Esfarjani *et al.* [PRB 84, 085204 (2011)] or Wang *et al.* [Scientific reports 4 6399 (2014)] according to which, phonons should have wavelength in 0.5 – 6 nm range. Other studies, for example Henry *et al.* [Journal of Computational and Theoretical Nanoscience 5(2) 141 (2008)], seem to predict even longer wavelengths at room temperature. Thus, while at 4 K we are quite certain about the phonon distribution and scattering mechanisms, at higher temperatures the situation becomes unclear. In any case, room temperature simulations require to perform first principle calculations not only to obtain the phonon distribution, but also to describe the Umklapp scattering process. We agree that modelling at 300 K would provide deeper understanding of our experimental results, but at the moment we are not capable of performing such calculations.

2. On the account that the interfaces possess roughness in the samples, how valid/accurate is the use of TDTR technique? What possible variations or errors in measurements owing to diffuse scattering of beams be present in the experimental readings of decay times? What is the thermal conductivity of these nanostructures on the basis of these measurements of decay times? The trends of thermal conductivity should be contrasted against existing works.

TDTR measurements give a global view of heat dissipation through the structures. If surface roughness increases, the decay time increases correspondingly. Currently, we are preparing a paper about the influence of surface roughness in

phononic crystals. In short, surface roughness causes lower thermal conductivity and this effect can be measured with our TDTR setup.

The values of thermal conductivity, extracted for aligned and staggered samples, are presented in Figure S3 of the Supplementary Information and are compared to the literature in Figure S5 with corresponding discussion. There seems to be good agreement with literature, as shown in the figure below. Please refer to the Supplementary Information for more details.

3. On Page 13, the possibility of thermal rectification is presented at the end of the main text of the manuscript. Since it is an important and interesting application, I would recommend that, on the very least, a cursory analysis on the degree of rectification in the suggested devices be presented to ascertain the future viability of these thermal lenses for the suggested purpose. Similarly, in conclusions and outlook, the authors discuss a contrast with conventional approach for hot spot creation without quantification and/or reasoning as to why their suggested method would be better.

At the moment, we are unable to quantify possible rectification, but plan to devote our next work to this matter. So, as this conclusion is perhaps too preliminary, we decided to omit mentions of thermal rectification in conclusions and the abstract, and only left one paragraph before the conclusion part.

As for the comparison with conventional photothermal approach, we meant that the advantage is that our lenses can potentially focus heat without a laser being focused on that spot and that the temperature can be controlled by the design of the lens (neck, number of rows). To clarify this idea, we rewrote the sentence as: “*Since such passive local heating does not require any external laser irradiation, it could become an alternative to the conventional photothermal approach. Moreover, the temperature at the hot-spot can be controlled via design of the lens, thus several hot-spots with different temperatures can be created on the same wafer.*”

4. In the second part of study (particularly on Page 10), the impact of surface features of the nanowire is expected to play a role in thermal transport and the effects of existing roughness in the samples should be included in the discussion. Additionally, even though the focus of the work is not explicitly on surfaces, the authors should be careful that the expression for specularity used for analysis is limited in its scope and is valid only for absence of

correlation. For the sake of completeness on the details about specularly particularly in context of thermal transport, I would recommend referencing studies where the surface correlations are included.

To address the impact of surface roughness, we have plotted the specularly parameter as a function of surface roughness and wrote a corresponding discussion including the role of surface correlations. As this discussion is rather lengthy, we placed it in Supplementary Information. Please refer to Figure S7 and the corresponding discussion.

5. The authors in the caption to figure 6 state that roughness is less than 5nm while in the text they state that roughness is 4nm. Additionally throughout the main text, it has been reported that $\eta \leq 2\text{nm}$ and the same value is considered for analysis. This should be clarified to avoid confusion.

To clarify, we changed the mini scale bars to 4 nm, and changed the text as “*we estimated that the peak-to-peak surface irregularity does not exceed 4 nm. Since the rms value of the roughness is less than its maximum amplitude, we assume $\eta \leq 2 \text{ nm}$* ”.

6. Figure 2 caption contains references to plots f-d (it should say e-h?). Are the error bars smaller than data points wherever they are not plotted in Figure 2 including e-h?

Thank you, the caption was corrected. To make the error bars more visible, we increased the cap sizes so that they go beyond the points.

7. At some places the choice of language should be reconsidered to avoid confusion. For example in the second paragraph it seems what the authors are suggesting is that the sole reason that the directionality of flux exists is the diffusive nature of phonons (irrespective of the direction of thermal gradient, entropy change etc). Another term that is used is ‘directional fluxes’. Since this term is not standard in the field. it should be explained/defined before being used in the text.

We have simplified the sentence as: “*practical application of the ballistic phonon transport remains challenging as different phonons travel in different directions.*”. We also corrected all the text to simplify understanding and avoid the uncommon term ‘directional fluxes’. For example, in the abstract we changed the sentence with “directional fluxes” into: “*First, we demonstrate that the arrays of holes form fluxes of phonons oriented in the same direction.*”, or in the introduction: “*we show that silicon phononic crystals orient ballistic phonons in the same direction and that this effect can be used...*”, and similarly in other places.

I recommend that the authors submit a revised manuscript addressing all the above concerns.

Reviewer #2

This manuscript deals with phonon transport in the ballistic regime. There are two key ideas: (i) the first one is to show that in the ballistic regime it is possible to shape the directionality of heat conduction with additional degrees of freedom in comparison to usual macroscopic diffusion, because there is no requirement of local thermal equilibrium and therefore no local isotropy. (ii) The second idea is that a hot spot can be generated passively by means of a “thermal lens”. This is in contrast to the usual knowledge stating that heat flows from hot to cold areas, and that hot spots are consequently required to be located only at the heat sources’ locations.

To demonstrate these ideas, three types of structures based on suspended membranes of 145 nm thickness have been fabricated and characterized by optical means in the transient regime: (i) 2D phononic crystals with periodically aligned or periodically disaligned holes, (ii) similar crystals connected to an array of nanowires, and (iii) “thermal lenses” connected to slits. These structures have been fabricated with up-to-date nanofabrication means. The experimental results are supported by Monte-Carlo simulations of phonon transport, probably the only way to describe heat transfer in the ballistic regime for some of the complex structures involved. However, the proof of the creation of a hot spot in-between the hot and cold baths by means of the “lens” stays indirect.

The work is very timely and really novel. This group has produced very interesting results in the last years and seems this time to bring the thermal transport community a new step forward. As a result, this manuscript can

certainly deserve publication in Nature Communications if the following questions can be answered in a satisfactory manner and if the comments are taken into account.

Questions and comments

- On the experiments and their analysis

1) No wave character of the phonons is taken into account in the analysis, while at 4 K the same group has previously claimed to have observed some effects. How do these results compare with the previous work: is the disorder previously introduced comparable with the disalignment here considered? A comment is expected in the manuscript.

Thank you for your interest in the activity of our group and for your high evaluation of our work. We think that, from the wave point of view, the staggered lattice should be regarded as an ordered lattice because this lattice is as strictly periodic as the aligned one, so coherence should be similarly present or absent in both; we also calculated phonon dispersions and both lattices have equally flattened branches. As far as a disordered lattice is concerned, from the particle point of view, the direct passages between the holes remains open in the 6% disordered lattice (in which coherence is already gone), whereas in staggered lattice the passages are closed; hence the increase in decay time in the staggered lattice, but only slight change in the disordered lattice (in incoherent regime).

However, the coherence effect is rather weak (10 – 20%) at 4 K and disappears above 10 K. For these reasons, we think that coherent effects can hardly affect data in the present work, but the effects of this work do affect the results in that unpublished work and this issue is addressed there. The figure below shows that Monte-Carlo simulations predict a slight increase in decay time when disorder is increased in 2D phononic crystals. This increase in decay time happens exactly because the passages between the holes start closing with disorder, so the ballistic transport decays. However, even in strongly disordered lattice, the passages remain rather open.

To clarify the lattice type, we have added one comment on pages 4-5: “Note that the staggered lattice should not be confused with a hexagonal (triangular) lattice, in which the density of holes is higher, or with a slightly disordered lattice, in which passages between holes remain open.”

And to discuss the coherent effects, we added on page 9: “Additionally, at low temperatures, some low-frequency phonons can be reflected specularly at all incidence angles and thus experience interference due to the periodicity of holes. At 4 K, for example, this effect can suppress thermal conductivity by 10 – 20 %. However, this phenomenon impacts aligned and staggered structures alike, as both are strictly periodic, thus it does not affect the observed difference between the lattices.”

2) It is stated that the increase in temperature due to the pump is less than 5 K. At $T_0=4$ K, an increase of $\Delta T=5$ K leads to a value of $\Delta T/T_0 > 100\%$, a situation where rectification effects are expected to be very strong. Such effects are in particular possible in the case of the “lens”, which is anisotropic. How could such effects, not necessarily due to ballistic transport, be discarded? Note in particular that it would have been more convincing if reference experiments with lenses in both directions without slits would have been shown.

Indeed, the relative increase in temperature might be large at 4 K, but this is true at the hottest point in the structure, and at the hottest instant. However, the results at 30 – 50 K ($T/T_0 < 100\%$) are nearly the same as the results at 4 K, so probably nothing exceptionally strong affects the results at 4 K.

Note, that absolute values of decay time here are not important because they depend on various factors. For example, absolute values of decay times for diverging lenses are different from those for converging lenses. This is because in converging lens heat from the metal pad has only five channels to enter the lens, so the decay times are longer, yet it does not prove rectification. For the same reason, the proposed experiments (without slits) would certainly yield different decay times, but it would not be an evidence of thermal rectification. To show thermal rectification using TDTR, we need to design much more complicated experiments. We plan to focus our next work to such rectification effects and we will try to address these concerns there.

3) In Fig. 4(d), the simulated lines are quite different from the almost linear trend found in the experiment. What could explain this difference? Is there a way to fit the experimental data with an improved model?

The simulated curve flattens as the length of the nanowires increases because we have a transition from ballistic to diffusive transport regime. The fact that the simulations are 2D accelerate this transition because there is no thickness.

We improved the model to simulate the phonons trajectories in 3D. As the phonons can now move in all directions, longer nanowire length is needed to reach the diffusive regime. As a result, the new results are in better agreement with the experimental points.

4) In the Suppl. Mat. It is stated that “each data points represents an average between several measurements”. Was actually the value fixed for a certain type of structure (then which one?), or in the same Figure two points can be average over a different number of measurements?

The number of measurements was fixed for each plot. For the aligned/staggered study, each room temperature point is the average of four measurements for the periods of 500 and 350 nm, three measurements for 200 nm. The only exception is the small structures of 160 nm in period, which break with time as the structure is very fragile. On these structures only four points are the average of three measurements, while others show values of a single measurement. So the typical error bars were assigned in that case.

For the coupled/uncoupled study, there were two measurements. For converging lenses – three measurements, for diverging lenses – two measurements and for the reference sample without a lens – four measurements.

By measurement here we mean independent measurements on different days or samples. Each measurement itself is an average of 30 automatic measurements.

5) Figs. 2(e-h), 4(d-e) and S3. The error bars should be set everywhere.

In these panels, the error bars are actually set everywhere, but they typically remain smaller than the size of most points. To make them more visible we increased the cap size.

6) The determination of the surface roughness over a square of 10×10 nm² is a bit limited. Usually larger areas are considered. In addition, what is the curvature radius expected for the AFM tip that was used?

We used the PRC-DF40P Olympus cantilever with a tip radius of 20 nm. We measured a larger area (14 x 50 nm), shown below, but showed smaller area in the manuscript for clarity.

7) Simulations in the diffusive regime for the cases of the phononic crystals were performed but not for the lenses with the slits. This is missing (can be put in the Suppl. Mat.) to be even more convincing.

The Comsol simulations for only phononic crystals (we also added the nanowire-coupled samples) were performed because they are rather simple systems. The lenses are more complex. For example, in the models we have to assign different thermal conductivity to different regions. Whereas in phononic crystal models the entire region has the same thermal conductivity, to simulate the lenses, we should assign thermal conductivities to the nanowires, the slit, the region with holes, and most importantly, we should probably assign different conductivities to the regions of different holes. Unfortunately, we found that the results of simulations in lenses are dependent on these parameters. For this reason, we decided not to publish such uncertain results. Instead, in the new version, we simulated the decay time experiments on the lens structures using our Monte-Carlo code and found an agreement with our experimental results.

- On the Monte-Carlo simulations

8) The Monte-Carlo simulations are apparently producing results in a stationary regime (stationary local energy distribution) while the experiments are transient. It is therefore biased. One could in particular imagine that ballistic phonons travel faster than those which are scattered and generate some diffusive part. Such mechanism is not mentioned anywhere. Some comments are expected here.

The Monte-Carlo simulations were performed in the stationary regime because the heat focusing effect could probably find applications in this regime.

The question of the bias of the ballistic phonons travelling faster than the diffuse ones is answered by the flat energy transmission as function of frequency as shown below. If ballistic phonons travel faster, it would mean that the transmission in low frequencies at stationary regime would be lower (because they already escaped the system). However, no such effect is observed. The fluctuations at low frequencies depict the lack of statistics because of the low number of phonons below ~800 MHz

The point raised by the referee is very interesting, and we decided to improve our code to simulate the decay times as those obtained experimentally. Today, it is not technically possible for us to reproduce the experimental conditions because of the amount of energy generated by the lasers is beyond reasonable calculation time (more than one year with our current capabilities). Nevertheless, the amount of energy can be reduced so that each simulation point takes a few days. Then, it is possible to compare the relative decay times between the measurements and the simulations.

We find a fair agreement for all the cases of the heat focusing study. The simulated decay times without the lens structure as well as the diverging lens show no significant dependence of the slit position. In case of converging lens, the simulations correctly predict the increase of the decay time when the slit moves away from the focus point. These Monte-Carlo studies in transient regime are now incorporated in the manuscript.

9) The Monte-Carlo simulations are performed in 2D, while the thermal transport is 3D. In a suspended membrane the effective average mean free path is reduced in comparison to a real 2D material (because of cross-plane motion): how was this taken into account?

The assumption was that the top and bottom surfaces were atomically flat, so they were not taken into account. However, we have improved our Monte-Carlo code to simulate the phonon motion in 3D structures. Now, we assume the top and bottom roughness of 0.3 nm, in accordance with the AFM measurements (Fig. 6a). All the simulations in this work have been replaced by the results in the 3D structures, including the transient studies in the previous question. The results, however, remain the same. Compare, for example, two figures below:

10) The roughness effect at the top and bottom surfaces of the suspended membranes is neglected. In reality there is certainly an eta value limited to that of the interatomic distance. This may reduce at least partly the ballisticity of the phonon transport. Is this taken into account?

Indeed, the top and bottom surface roughness reduces the phonon mean free path. Our new 3D simulations take this roughness into account ($\eta = 0.3$ nm), yet this roughness does not seem to significantly affect our discussions and the observed effects, as shown in the answer to the previous question.

11) For the simulations $\lambda = 3$ nm seems large at room temperature. It is probably closer to 1.5 or 2 nm (maybe even lower), and this has an effect on the specular parameters. How do the results change for smaller wavelength?

We use the first principle calculations by Esfarjani et al. [PRB 84, 085204 (2011)] as the reference for the room temperature phonon wavelengths. They predict phonon wavelengths in Si to be in the 0.5 – 6 nm range. However, for the discussion in page 8, we changed the dominant wavelength to 2 nm, which basically does not affect the conclusion that phonons approaching the surfaces tangentially ($\theta > 80^\circ$) have specularity higher than zero. Results for other wavelengths were added to Supplementary Information. However, this is just a quick estimation from Soffer's equation. For the Monte-Carlo simulations we used the full spectrum of phonons given by the Plank distribution.

12) In Fig. 3, only ten rows are simulated, so that the structure mostly acts as a filter (the mean free path is larger than the full size of the structure) and there is no effective diffusion regime (as in nanowires or suspended membranes that are longer). What is actually the average mean free path at this temperature?

Although we didn't find any data precisely at 4 K, the mean free path at this temperature is probably longer than 10 μm [Regner *et al.* Nat. Comm. 4 1640 (2013); Minnich *et al.* PRL 107 095901 (2011)]. Hence, the phonons propagating between the holes can probably traverse these ten rows ballistically, even if some specular scattering events might occur on the holes.

13) It is stated on p.11 that to model the transport in the nanowires Lorentzian and uniform (I guess cosine?) distributions of initial angles are considered. Why not accounting for the results of Fig. 3?

The Lorentzian shape comes from the analysis of Fig. 3d for which the peak can be well fitted by a Lorentzian curve. The initial angle of the phonons entering the nanowires are randomly picked in this distribution. More technically, the random package of the C++11 version contains a Cauchy (Lorentzian) distribution generator. It was combined with the Mersenne Twister mt19937-64 random number generator [ACM Trans. Mod. Comp. Sim., 8, 3 (1998)] to draw the corresponding angle distribution.

14) How are Normal vs Umklapp processes accounted for in the simulations?

In our simulation we take into account only the normal process as the probability of Umklapp process is negligibly small at 4 K. In our algorithm, at each time step, each phonon can be scattered in random direction with a probability of $1 - \exp(-t/\tau)$, where t is the time passed since the previous internal scattering event of this phonon and τ is characteristic internal scattering time given by $\tau^{-1} = \tau_{\text{impurity}}^{-1} + \tau_{\text{normal}}^{-1}$, with $\tau_{\text{impurity}}^{-1} = 2.95 \times 10^{-45} \omega^4$ and $\tau_{\text{normal}}^{-1} = (2\tau_{\text{TA}}^{-1} + \tau_{\text{LA}}^{-1})/3$, where $\tau_{\text{TA}}^{-1} = 9.3 \times 10^{-13} \omega T^4$ and $\tau_{\text{LA}}^{-1} = 2.0 \times 10^{-24} \omega^2 T^3$.

- Completing and improving the manuscript

15) All the simulated local energy maps are given with arbitrary units, while it would be very interesting to know the local effective temperatures, in particular the one of the hot spot in comparison to that of the baths. I suggest to add the units in Figs. 3 (a-d) and 5 (b,d).

The question of the definition of temperature at the nanoscale is still a debated topic in the community. For this reason, we prefer to show directly the simulation output which is given in terms of energy. At 4 K, we could use the direct relation $E \sim T^4$ between energy and temperature to give an equivalent temperature. Should Reviewer insist on the thermal map, we could replot this equivalent temperature, but we should stress that it should not be interpreted as the real temperature.

16) I suggest adding a figure (at least in the Suppl. Mat.) with the value of the specularity factor p as a function of temperature for a given value of roughness η : this would help the reader to understand the effect of temperature.

We agree with the suggestion. To help the reader understand the effects discussed in our work, we have added the figure below and corresponding discussion to the Supplementary Information. In this figure, the reader can see the range of incidence angles in which phonons remain specular (which is important for the discussion in the main text), whereas wavelengths of 25 and 2 nm approximately correspond to the dominant phonon wavelengths at 4 and 300 K, respectively, so the impact of the temperature is also covered.

17) It would be good to have also a picture (not a schematic) of the full device (with Al pad and baths).

We have added SEM images of devices of different kinds in the Supplementary Information.

18) On p.9, it is written that heat transport is ballistic in nanowires over few micrometers in length. Please give more details here (material, temperature considered) as it is for example not the case for silicon nanowires at room temperature.

Indeed, it is not the case in silicon at room temperatures. Recently, we have investigated this effect in silicon nanowires at different temperatures. Figure below shows that there is a clear sign of ballisticsity at 4 K, but nothing at room temperature.

Currently, the paper about this effect is resubmitted after a revision and hopefully will be published soon. For now, we corrected the text as: “...being mostly ballistic in silicon nanowires of few micrometres in length, at low temperatures⁴⁹, but becoming diffusive as the length is increased.”, where Ref. 49 is the PhD thesis “Thermal phonon transport in silicon nanostructures” by Jeremie Maire. If the paper is accepted soon, we will change the reference.

19) Fig. 4(e): Strange to have an x-axis that starts below 0 K...

We set a small margin just to make the first point visible, and reduced even more in the new version.

20) In the introduction I suppose that the authors refer to “in-plane transport” and not “cross-plane transport” when citing Ref. [9].

In that work, the authors use 2D phononic crystals, but they actually study the heat conduction in the cross-plane direction.

21) Some of the prospects in the last section might be softened by changing “can” into “could”...

We agree and changed accordingly.

22) A reference is expected for the correction factor F in the Suppl. Mat.

To explain the choice of correction factor, we performed additional analysis. We obtained effective thermal conductivities for all structures using the same FEM modeling, but without holes in the membranes. Next, we compared these effective conductivities with those obtained by simply using this correction factor. In the figure below we can see that they are in a good agreement.

We have also added a couple of references supporting the factor F in the form we use.

23) At many place "heat transport" is written. If "phonon" is not mentioned, I suggest to replace "transport" by "conduction" since other means of thermal transport exist (in particular radiation).

We agree and changed accordingly.

- Errors and typos:

Note that there are still some English language mistakes throughout the manuscript. Among errors and typos, one can find:

- (a) Fig. 2: it is (e-h) and not (f-d)
- (b) Fig. 2 (2x) and Fig. S3 (2C): It is "diameter-to period" and not "dimeter-to-period".
- (c) p.11: "simulations predict" and not "predicts"

Thank you, the errors have been corrected.

Reviewer #3

The authors study thermal transport by phonons in a series of nanofabricated silicon thin films. There are three types of structure: (i) periodic pore arrays (i.e., phononic crystals), which either allow or block a direct line of sight for phonon transport, (ii) phononic crystals with nanowires attached to one end, which either allow or block a direct line of sight for phonons to enter the wires, and (iii) holes of different sizes positioned to focus phonons. In their TDTR-based experiment, a laser heats a metal pad that is in lateral contact with the nanofabricated films. A second laser monitors the temperature response of the pad as thermal energy dissipates through the film. A decay time is extracted and used to compare the structures. Complementary Monte Carlo and continuum simulations are used to interpret the data.

This study is highly original, well conceived, and carefully thought out. The authors have considered structures more complicated than previous works. They do a very good job in explaining their results in a thoughtful, logical manner. While none of the results are highly surprising, the fact that the authors were able to build structures where clear evidence of ballistic transport and focusing are present is an outstanding achievement.

Comments

1. The comparison between structures is based on the decay time extracted from the TDTR experiments. While I am not arguing that this single number does not resolve important differences between the samples, there is a lot going in these experiments, where there is a full spectrum of phonon modes. Can the authors comment on what other information might be extracted from their experiments (now or in the future) that would be useful in

providing more insight? As of now, there is no way to directly compare the experimental measurements to the simulations, which might provide more insight into the underlying phonon physics. One idea is to analytically solve the heat diffusion equation in one dimension in the samples (with an effective thermal conductivity) and see what time scale emerges.

Thank you for your interest in our experimental technique and for the high evaluation of this work. There are few hypothetical ways to extract more information using our experimental setup. Firstly, from year to year we struggle to increase the sensitivity of our system in order to decrease the excitation power. Although we currently don't see any dependence of the decay time on the excitation power (at least for reasonably low power) even at low temperatures, we believe that at very low excitation powers the decay should become faster, and dependence on excitation power would give us more insight. Moreover, if we could use much lower excitation powers, we could directly compare Monte-Carlo simulations and experimental results, whereas we are currently unable, of course, to simulate the real number of phonons generated by the power of 300 nW.

Secondly, we recently recognized that we can study excitation time as well as the decay time; that is, when the excitation pulse is long enough (several μs), how quickly the signal saturates. We found that this quantity differs from one sample to another. However, we are still not completely sure how to interpret these data.

2. Will the decay always be exponential? What governs that behavior?

The in-plane decay should be, and experimentally always is, exponential. The solution of the thermal diffusion equation can be reduced [Luckyanova *et al.* Nano Lett. 2013, 13, 3973] to $T = T_0 \cdot C \cdot (\alpha_z t)^{-1/2} \cdot \exp(-\alpha_x q^2 t)$, where α_z and α_x are cross-plane and in-plane thermal diffusivities, t is time and C is a constant. As long as only in-plane heat conduction is concerned the change in temperature (T) is proportional to the exponent. We have added this mention to the methods section.

3. What exactly is the “thermal energy distribution” plotted in Fig. 3a?

The energy distribution maps are obtained as follows: once steady state is reached, the algorithm starts recording the coordinates and energy $\hbar\omega$ of phonons; the energy is integrated over 10 000 time steps (10 ns) being recorded at every time step for each phonon into the pixel corresponding to the in-plane coordinates the phonon. Thus at the end we have integrated values of energy for every pixel of the map. We have added this explanation to the Methods.

For the spatial distribution plot (on top of the map), the width of the membrane was divided into 360 segments. In steady state, each time a phonon exits the phononic crystal, its energy is recorded in the corresponding segment. The spatial energy distribution is the final account of the energy of each segment. To obtain the angular distribution, the same procedure is used by segmenting angles instead of position.

4. Page 8 and related discussion on page 9. The authors specify one phonon wavelength in the discussion of the boundary scattering. It would be better to call this quantity the dominant phonon wavelength and acknowledge that (i) there is a range of value at any temperature, and (ii) this range moves to larger wavelengths as temperature decreases due to the Bose-Einstein occupation.

We agree and corrected the discussion as “*The range of phonon wavelengths depends on temperature and shortens from 10 – 100 nm at 4 K to 0.5 – 6 nm^{38,39} (0.5 – 50 nm⁴⁰) at room temperature. Assuming, for instance, the dominant wavelength of $\lambda = 2$ nm and $\eta = 2$ nm (Methods), we can estimate that...*”.

5. In analyzing the nanowire structures, the effect of the phononic crystal is included in tau, which may decrease the measured difference between the coupled and uncoupled cases. Can the authors conceive of a way to better isolate the nanowire effect?

Indeed, absolute values of the decay time contain information about the phononic crystal part (and probably the heat sink) and can decrease the measured difference. For this reason, we tried to minimize this effect and have chosen the minimum number (ten) of hole rows which is sufficient to create strong directionality. To estimate how these ten rows affect the decay time, we performed an experiment where the length of the bridge was fixed and we gradually added rows of holes next to the metal pad. The figure below shows that, as we add 10 rows of holes to the membrane (as it is in our samples), we increase the decay time by about 30%.

In an attempt to isolate the nanowire effect, we also tried a FEM analysis: using known values of thermal conductivity of phononic crystals (PnC) and nanowires (NWs), we used the heat transfer module in Comsol Multiphysics to simulate the same experiment (as explained in Supplementary Information). First, we simulated PnC + NWs, next only PnC and finally only NWs. For the PnC + NWs configuration we obtained the decay time 1.178 (2.293) μs for 1 (6) μm long NWs, whereas for only PnC and only 1 (6) μm long NWs we obtained 0.974 and 0.312 (1.372) μs , respectively. So the decay time of PnC + NW structure is nearly (<10% difference) equal to the sum of decay times of PnC and NW structures. The longer the wires the smaller is the error due to the impact of the heat sink. Thus, perhaps, to exclude the impact of the PnC part from our experimental values we could just remove certain part of the decay time (namely the 0.974 μs).

Such approach, however, would make the discussion much more complex, and the data will probably become less accessible for a wide readership. More importantly, there is no guarantee that such approach is totally correct. For these reasons, we decided to keep the direct experimental data for now.

6. I do not understand the wire simulations (page 11). Are only the wires being considered or is the phonic crystal part of the simulation as well? More detail would be useful, maybe in the Methods section.

As it is difficult to build Monte-Carlo models for too complex geometries, the simulation of the nanowire-coupled samples was conducted in two steps: first, we simulated only 10 rows of phononic crystal and obtained distribution of the phonon exit angles after these 10 rows of holes (Figure 3); next, we simulated only a single wire, where phonons at the entrance were given the initial angles approximated from distribution of exit angles obtained on the previous step: phonons in the wires attached in the coupled configuration have Lorentzian distribution of initial angles, whereas phonons in the wires attached in the uncoupled configuration have uniform distribution. We have added these details to the Methods section.

7. Page 11: “less than half of the phonons can traverse ballistically even in the 1 micron long wires.” What is the basis for this statement?

To clarify, we changed the reference at the end of this sentence to [Esfarjani *et al.* PRB 84, 085204 (2011)]. In this work, authors plot the phonon MFP spectrum for Si at 300 K (Fig. 6); from this plot we can see that more than half of the phonons have MFP < 1 μm , and thus cannot traverse 1- μm -long nanowires without experiencing internal scattering.

8. Page 16. The phonon model used is basic. Are the phonon lifetime relations valid at all temperature considered? Given that the comparison between the experiments and simulations is qualitative, I am not particularly concerned.

Our model is probably only valid below 20 K, because of the Debye approximation. For this reason, all the simulations are performed at 4 K.

9. SI page 1. The thermal conductivities of the silicon and aluminum thin films are taken to be 75 and 237 W/m-K. How were these values obtained?

The value of $75 \text{ Wm}^{-1}\text{K}^{-1}$ for the silicon corresponds to the value of thermal conductivity of the suspended thin film of thickness 145 nm. This value was obtained in our previous work and agrees with the literature [Anufriev *et al*, PRB 93 045410 (2016)]. The value for aluminum is the commonly used bulk value, as we were not able to measure the value for the thickness we deposited. Although aluminum thin films probably have lower thermal conductivities than the bulk, the value used here has a negligible impact on the extracted value of thermal conductivity given the time scales involved. We have added the references to these values in the Supplementary Information.

10. SI page 4. The $F(\phi)$ factor is geometry dependent. I suggest that the authors not provide any form of this equation unless there are explicit about what geometry it applies to.

To explain the choice of correction factor, we performed additional analysis. We obtained effective thermal conductivities for all structures using the same FEM modeling, but without holes in the membranes. Next, we compared these effective conductivities with those obtained by simply using this correction factor. In the figure below we can see that they are in a good agreement.

We have also precised that the correction factor in this form applies to porous geometries and added couple of references to the form we use.

11. SI page 5 "... have escaped from the phononic crystal during the transient regime." In the Methods section, there is discussion about how data was collected from the simulations once steady state was reached. As such, why would the transient part of the simulation be relevant here?

Thank you for pointing out this detail. We reconsidered this matter and realized that our explanation of those small dips in the peaks was wrong. Indeed, it has nothing to do with the transient part. The real reason for these small dips in the peaks is simply statistical: the probability that phonon have close to zero angle is defined by initial uniform distribution, but probability that phonons have some other angles is slightly higher, because phonons that ended up with such angles due to the scattering on holes are added to those that initially had such angles. We have corrected this explanation in the Supplementary Information.

List of main changes

- 1) Monte-Carlo simulations were improved and became three dimensional. Thus, all the theoretical data in the manuscript were replaced by new simulations results. However, there is almost no visible change.
- 2) Theoretical curves are added to Figs. 5e-g; the agreement is discussed in the text.
- 3) We improved general clarity of the text. Specifically, according to Reviewers' suggestions:
 - a) Added a paragraph about coherent effects.
 - b) Precised phonon wavelength at different temperatures.
 - c) Replaced various ambiguous terms, phrases and values.
- 4) We added details on Monte-Carlo simulations to the Methods section according to Reviewers' questions.
- 5) We significantly extended Supplementary Information according to Reviewers' requests by adding:
 - a) Plots and discussions on values of thermal conductivity obtained in this work and the literature.
 - b) Plots and discussions of specular parameter and its dependence on surface roughness (and its correlation), phonon wavelength (temperature).
 - c) SEM images of full devices
 - d) Details of Monte-Carlo simulations: (i) general algorithm and (ii) decay time simulations.

Reviewers' comments:

Reviewer #1 (Remarks to the Author):

In the revised version of the manuscript, the authors have addressed some important concerns previously raised to a large extent. One of the main concerns that remains unaddressed is expanded upon in Point#2. This should either be resolved or an explanation be included in the manuscript.

1. From my understanding of Fig. S6 and S7, some low frequency phonons would be reflected specularly irrespective of temperature and the relative proportion of heat carried by them would change according to temperature. The phrasing "Additionally, at low temperatures, some low-frequency phonons can be reflected specularly at all incidence angles..." should be reconsidered to avoid ambiguity.

2. In my opinion, the variability of the phonon wavelength spectrum and Umklapp relaxation times is not an issue if the final desired outcome is an analysis of the ballistic transport at room temperature. The authors could even compare their results using different available phonon wavelength spectra (Esfarjani et al., Henry et al. etc) and see the differences arising from using varied phonon spectra. Since the authors have accounted for a spectra at low temperatures (line 158, 10nm-100nm), the exact value of the spectra range should not be a deterrent for such a critical analysis. The authors have stated that their unwillingness to bring forward such an important analysis relevant to their most significant conclusion is partially hinged upon the fact that the said analysis "may raise more questions than answers" while in fact raising such questions would indeed advance the research in nanoscale thermal transport. Additionally, the figures and discussion in its current form are inconsistent. [Figure 2, e-i shows the data for $T=300\text{K}$ and is used to construct a hypothesis for ballistic transport for a range of temperature (30-300K) based on the fact that the difference in decay times strengthens at 30K (Line 97). However, computations of Figure 3 is for $T=4\text{K}$ and is used to explain the hypothesis for the same range (30K-300K).]

All the above points should either be resolved or an explanation be included in the manuscript.

Reviewer #2 (Remarks to the Author):

The authors have revised their manuscript and sent a lengthy rebuttal letter. The original manuscript was already in a good shape and the work has been improved on few points. I summarize the few last points to tackle at the end of this (also lengthy!) second-round review.

The following remaining weaknesses have been found by reading the current version of the manuscript.

- A) There is no explanation of the effect porosity on the decay time.
- B) p.8: While it is right that reaching tangentially a surface allows for specular reflection, one should keep in mind that the probability of such a large angle is usually very small.
- C) Fig. 4d: What could explain that the required roughness increases when the length increases? In principle the roughness does not change with length.
- D) Fig. 4e: Why there is no simulation for the temperature dependence? Was there something strange there in the simulated results there?
- E) Line 159: mentioning 50 nm at room temperature from the manuscript by Henry and Chen is

too large as the contributions of these wavelengths seem very small (less than 5%).

F) Fig. 5(e-g): The definition of quantity of the ordinate axes is not given in the core of the manuscript. Please indicate it.

G) Line 244: Please mention the criterion used for determining the size of the hot spot in the text also: the full width at half maximum, if I understood correctly.

H) Line 187: replace "it does" by "it should" (no evidence, only a supposition)

I) Line 278: "could" instead of "can": only supposition.

J) Line 98: typo, remove "the" in "the most data".

K) Typos everywhere (at least 5x in the manuscript and in the Supplementary Material): Planck, with a "c". Example: Lines 351

L) References: Please harmonize, either writing all the author names or first author "et al." The same comment applies for the Supplementary Material.

The following remaining weaknesses have been found in the current version of the Supplementary Material

S.A) Monte Carlo simulations. When a surface emits phonons isotropically, as is the case for a blackbody in radiation, the random angle theta is usually drawn according to a cosine distribution for phonon bundles. Here it is apparently drawn without this cosine distribution, so that more phonon bundles are launched with large angles than it should. This is a strong issue, as it fastens the heat dissipation from the aluminium surface. The authors should correct their simulations or prove that this has no impact on the results. This is a key point.

S.B) The comment on the fact that it is not possible to reproduce numerically the exact decay time is very intriguing. I do not at all understand why the decay should not be the same depending on the power of the incident light if the heat equation in the linear regime is considered. This is a key point that may shake the whole demonstrations in the manuscript.

S.C) Fig. S5: Please highlight that the ordinate axis is in logarithmic scale, so that the data dispersion is not easily observable. There are variations by factors 2 or 3 for some values of the neck...

S.D) Fig. S11: the captions of the abscissa are cut.

S.E) Line 27: "non-Fourier" or "non-diffusive", but not "non-classical" as nothing is quantum here.

S.F) Line 30: Typo, remove "do"

S.G) Line 79: remove "most"

S.H) Lines 94-95: replace the erroneous sentence by "thus using Eq. S1 to estimate the specular parameter seems reasonable."

I now review the answers in the rebuttal letter.

Reviewer #1

R1.0. "Questions about thermal phonon coherence: why not accounted?"

The authors explain that there that staggered structures stay periodic and indicate that they found similar dispersion curves. It would be good to also provide these data to the reviewers so that this claim can be analyzed. In particular, the authors do not indicate up to which frequencies they computed the data.

R1.1. "Distribution of phonons not accounted"

The authors argue that the actual distribution is not known. The answer is not really satisfactory. Planck distribution is cut by the Debye wavelength, there is no need to consider lower wavelengths. This should actually provide already interesting results. It would be interesting to see if the results are maintained by considering the not-too-large wavelength, knowing that possible large wavelengths would make the effects stronger.

R1.2. "Roughness vs TDTR?": the question and the answer are not really useful.

R1.3. "If mention of rectification, then quantify it"

The authors have wisely erased the speculative mention of rectification, as it is not the message that is conveyed here.

R1.4. "Specularity and correlation"

A new section has been added, which is good. One could however argue that RIE induces correlation as was shown by SEM images in some papers in the past. But maybe the associated wavelength is too large.

R1.5. "Clarify roughness values": this has been done.

R1.6. "Errors in figure numbers": this has been corrected.

R1.7: "Directionality and "directional fluxes""

Contrarily to this reviewer, I was not shocked by the previous way it was written. The new one is also reasonable.

Reviewer #2

R2.1. "Wave character is not considered": the answer is satisfactory.

R2.2. "Large delta T in comparison to temperature at 4 K."

The authors indicate that despite this fact nothing special seems to appear in comparison to higher temperatures.

Another comment is given, stating that the decay times depend also on the geometry, not only on the thermal conductivity, and that inverting the geometry is sufficient to change the decay time even through staying in the linear regime. Such comment should be inserted at least in the Supplementary Information. I suppose that it has been observed in linear FEM simulations.

R2.3. "Simulations with the nanowires are not very well matching the experimental data": the authors state that 3D Monte Carlo now better matches.

R2.4. "How averaged are the data? Dispersion?"

How the data are averaged has been clarified in the rebuttal letter. It would be good to clarify that even better in the manuscript.

R2.5. "Error in labeling": this has been corrected.

R2.6. "How is the roughness measured?"

The answer is OK. Data on the tip radius of curvature should be given at least in the Suppl. Mat..

R2.7. "Why not simulating the lenses with FEM?"

The authors indicate that they would need to include thermal conductivity that locally depends on size/thickness/width of the structure and that it is difficult. They estimate that the Monte Carlo simulations are sufficient to find the trend. While I agree that the Monte Carlo simulations seem to show the same trend, the comment on the fact that even in the diffusive regime the decay time depends on the orientation leads to require these FEM simulations. Maybe they cannot be quantitative, but it would be important to see if they provide another type of trend or if the trend is the same than that of the Monte Carlo data. Indeed, if they have the same trend, then nothing is demonstrated...!

R2.8. "Simulations in the stationary regime and not in the transient one": the authors now perform transient simulations, so it seems now more reasonable.

Another comment on the transmission as a function of frequency is not really relevant but is not key to the work.

R2.9. "2D vs 3D Monte Carlo simulations": the authors have modified their simulations and now perform 3D simulations.

R2.10. "Roughness on membrane surfaces": it is now taken into account.

R2.11. "Main contributing wavelength of 3 nm seems too large": this has been modified.

R2.12. "Structure is too small so that it is a filter: there is no diffusive regime."

The authors know that the mean free path at 4 K is much larger than their structure size (in fact their expression for the relaxation time should help them to evaluate it!). It would be good to determine the effective mean free path in their structures by accounting diffuse scattering at boundaries. However, I verified and there is no determination of thermal conductivity at 4 K, thus there is no problem in the manuscript.

R2.13. "Why taking a Lorentzian curve and not the actual data?"

The authors considered that Fig. 3d can be fitted by a Lorentzian curve. One can see that the

bottom part will be certainly underestimated in this case, so the effect of the directionality might be artificially increased with this procedure...

The same question than S.A. asked higher about the cosine direction for the "emission" of phonons was not answered.

R2.14. "Normal vs Umklapp scattering in the simulations"

Only Normal scattering is considered because all the simulations were performed at 4 K. Avoiding simulations at 300 K is a weakness of the manuscript.

R2.15. "Local effective temperature"

I think that providing the local effective temperature (what will be useful to compare with what is known) is important. Of course the authors should explain that this is not a thermodynamic equilibrium temperature (clear from the non-isotropic temperature distribution!), but that this is just the local energy. By the way, the whole paper is based on anisotropic distribution function for phonons (so angular distribution of energy is plot), but the distribution in energy (maybe along some direction) is not shown.

Note that the local energy is not proportional to T^4 for phonons at room temperature.

R2.16. "Specularity factor as a function of temperature": this has been done.

R2.17. "Pictures of the devices": they have been added.

R2.18. "Heat transport is ballistic at micrometers": the new wording is reasonable. Note that the reference is not complete ("PhD thesis" missing).

R2.19. "Strange to have an axis below 0": Still the case. I leave this open to the authors.

R2.20. "Cross-plane vs in-plane studied in a reference": Yes, the authors are right.

R2.21. "Last section to be softened": OK.

R2.22. "Filling factor in the phononic crystals"

A new section has been added: it is clear now. However in the manuscript it could be good to always show that the effects are stronger than those expected from simple effective medium theory.

R2.23. "Heat conduction" and not "heat transport": this has been done.

Typos underlined are also corrected.

Reviewer #3

R3.1. "TDTR provides more information than what is used. Simulations difficult to compare with experiments. Analytical solution for heat diffusion possible?"

The authors do not directly answer to this not-so-clear question. For instance, the authors do not reproduce exactly numerically the experiments: the intensity is always normalized, which weakens the trust of the reader in the numerical reproducing of the data. Some improvement or comment is in order here in the manuscript.

R3.2. "Decay always exponential?"

The authors indicate that the decay is always exponential if the regime is diffusive, but do not tackle other regimes. This points to R2.12: if there is no diffusive regime, why should the decay be exponential? An answer can be that the decay will always be diffusive finally at some location, maybe far from the structure. Another answer is also that all observed decays seem exponential...

R3.3. "Thermal energy distribution"

The authors explain the procedure from the Monte-Carlo simulation, but it would be good to provide expressions as a function of the distribution function. This is easy. Writing this part will also make easier the answer to R2.15.

R3.4. "Broad distribution of phonon wavelengths": the answer of the authors is satisfactory.

R3.5. "Decay time considered includes both phononic crystal and nanowires": the authors have studied this effect experimentally and numerically. Since the goal is to see the coupling, I feel that what has been done is sufficient.

R3.6. "Not understanding the wire simulation": this is well explained.

R3.7. "More than half of the phonons have MFP larger than 1 micron": this is clear.

R3.8. "Phonon model is basic".

This is a good point from the referee. The authors indicate that they perform the simulations at low temperature to avoid issues at room temperature. While not having the simulations at room

temperature is a weakness, there is some logics in the manuscript.

R3.9. "Values of the thermal conductivities considered": this is well explained. If I understood correctly, these values are however not used at low temperature.

R3.10. "Geometrical factor is maybe geometry-dependent...": the text in the Supplementary Material is now useful.

R3.11. "Steady-state vs transient state"

The authors indicate that a previous explanation was incorrect, but that it is corrected in the current version.

To conclude, the main remaining points that need to be tackled are S.A, S.B, R2.2, R2.7, R2.15 (R3.3), R3.1, R3.2. The reviewer would also have preferred to see some data for R1.0.

Once all these points have been addressed, the manuscript could be published.

Reviewer #3 (Remarks to the Author):

I thank the authors for their careful consideration of my comments and those of the other reviewers. They have made their already strong manuscript even better. I recommend publication in Nature Communications.

Comments as related to my original numbering scheme

2. The temperature response provided by the authors in the rebuttal, as they themselves note, is based on a solution of the thermal diffusion equation. As such, it is inherently diffusive. The central focus of this work is on ballistic transport. As such, it seems that the exponential decay is an approximation, although apparently a good one.

4. The range of wavelengths reported in Ref. 40 (up to 50 nm) does not seem correct. Could it be due to the use of an empirical potential and not first principles? I suggest that the authors contact Asegun Henry (now at Georgia Tech) to ask his opinion.

5. I appreciate the extra analysis. While implicit, I think that it would be worth mentioning in the text that the tau for these structures includes the effect of the phononic crystal and the nanowires.

10. Thank you for providing this additional information in the SI. I note that Jain et al. (Ref. 27 of the main text) found that the Hashin factor, $(1-\phi)/(1+\phi)$, provided a good comparison to their COMSOL simulations of similar structures as considered here.

New Comments

i. Page 6, line 118. Better to say "effective thermal conductivity" due to the presence of ballistic effects.

ii. Page 8, line 155. When the authors write "diffuse scattering in random directions," what distribution is being sampled? It should be a Lambert cosine.

iii. Page 9, line 173. I suggest changing "strengthens" to "increases" to avoid ambiguity.

iv. Page 12, line 239. Word "however" is not appropriate here.

Reviewer #1:

In the revised version of the manuscript, the authors have addressed some important concerns previously raised to a large extent. One of the main concerns that remains unaddressed is expanded upon in Point#2. This should either be resolved or an explanation be included in the manuscript.

1. From my understanding of Fig. S6 and S7, some low frequency phonons would be reflected specularly irrespective of temperature and the relative proportion of heat carried by them would change according to temperature. The phrasing “Additionally, at low temperatures, some low-frequency phonons can be reflected specularly at all incidence angles...” should be reconsidered to avoid ambiguity.

We agree and corrected as: *“Additionally, some long-wavelength phonons can be reflected specularly at all incidence angles (Supplementary Information) and thus experience interference due to the periodicity of holes^{28,50}. Although at room temperature, such phonons represent only a small part of the phonon spectrum in our system; at low temperatures, their portion may become significant.”*

2. In my opinion, the variability of the phonon wavelength spectrum and Umklapp relaxation times is not an issue if the final desired outcome is an analysis of the ballistic transport at room temperature. The authors could even compare their results using different available phonon wavelength spectra (Esfarjani et al., Henry et al. etc) and see the differences arising from using varied phonon spectra. Since the authors have accounted for a spectra at low temperatures (line 158, 10nm-100nm), the exact value of the spectra range should not be a deterrent for such a critical analysis. The authors have stated that their unwillingness to bring forward such an important analysis relevant to their most significant conclusion is partially hinged upon the fact that the said analysis “may raise more questions than answers” while in fact raising such questions would indeed advance the research in nanoscale thermal transport. Additionally, the figures and discussion in its current form are inconsistent. [Figure2,e-i shows the data for T=300K and is used to construct a hypothesis for ballistic transport for a range of temperature (30-300K) based on the fact that the difference in decay times strengthens at 30K (Line 97). However, computations of Figure 3 is for T=4K and is used to explain the hypothesis for the same range (30K-300K).]

We would like to note that the main point of this work is the experimental demonstration of the directional heat flow. Whereas this effect is most visible at low temperatures, we suppose that the experimental results as a function of temperature can be interesting for the readers, so we included the data at higher temperatures. Although, we would like to also perform temperature-dependent simulations, the development of the code is too important to be accomplished in a few months, without accounting for intermediate steps that would be required to verify the results. Thus, while we continue to work on the development of the simulation, presently we would like to deliver the experimental findings to the community. We will appreciate if reviewer allows us to leave the temperature-dependent simulations as our future work. Since the effects strengthen at low temperatures, we focused our simulation on our lowest available temperature (4K).

To explain it, we added in the main text: *“Due to the Debye approximation used in our Monte Carlo algorithm, we conducted all simulations for the temperature of 4 K.”*

However, in this revision, we tried our best to provide more information to readers and to give some idea about changes that might be induced by temperature. We performed simulations for different dominant wavelengths so that in each simulation all phonons are emitted with a single wavelength. We chose to take four wavelengths between 25 nm and 2 nm that encompass the expected range between 4 K and 300 K. The results are shown below.

Phonons with shorter wavelengths experience more diffuse scattering events on the geometrical boundaries as well as more internal scattering events, increasing the decay time as shown in the panel a. By fitting the decay curves, we observe that the decay time constant decreases exponentially (panel b) as the wavelength increases.

Panels c-f show the energy maps near the focal point for phonons of different wavelength. Naturally, as the wavelength is reduced the hot spot becomes blurred until it is no more visible.

According to the dominant wavelength formula above with $v = 6000$ m/s, we can estimate the corresponding temperatures as $T(\lambda_d = 25nm) \approx 4K$, $T(\lambda_d = 10nm) \approx 10K$, $T(\lambda_d = 5nm) \approx 20K$ and $T(\lambda_d = 2nm) \approx 50K$. However, formula is valid only at low temperature as it is based on the Debye approximation [Ramiere *et al*, AIP Advances 7, 015017 (2017)]. As we previously discussed, the wavelengths at room temperature are estimated to be actually at least in the 0.5 – 5 nm range. As we can still observe the hotspot when $\lambda = 5$ nm, it may explain that a small dependence on the slit position remains up to 300 K in the inset of Figure 5g in the main text.

Finally, although we cannot use this analysis to establish a temperature dependence, it shows the correct tendency of the reduction of the directionality effect with the decrease of the phonons wavelength.

We added these graphs and brief discussion of the impact of the wavelength to Supplementary Note 7.

We also added in the main text: *“In the Monte Carlo simulations, we also observed that the hot spot gradually disappears as phonon wavelengths are increased (Supplementary Note 7). However, since the simulations are strictly valid only below 20 K, the temperature dependent simulations should become the future work.”*

Finally, about the seeming inconsistency between temperatures of simulations (4 K) and the data it aims to explain (30 K). We decided to show 4 K simulations, because Fig. 3 is a bridge between aligned/staggered experiment (at 30 K) and nanowire-coupling experiment (at 4 K). We could have present 30 K and 4 K simulations for both experiments, correspondingly, but the figures would be quite similar. Note that in the simulations in Fig. 3 we also already use dimensions of the structure from nanowire-coupled experiment, in order to use Fig. 3 to explain the data in Fig. 4 and even use the angle profiles for the simulations in Fig. 4. Thus, it is mostly to facilitate the reading, we decided to show the simulation of only one structure at one temperature to explain both types of experiments and smooth the transition between them.

Reviewer #2:

The authors have revised their manuscript and sent a lengthy rebuttal letter. The original manuscript was already in a good shape and the work has been improved on few points. I summarize the few last points to tackle at the end of this (also lengthy!) second-round review.

The following remaining weaknesses have been found by reading the current version of the manuscript.

A) There is no explanation of the effect porosity on the decay time.

We added an introductory passage as the beginning at the experimental results: “Generally, heat dissipation becomes slower as diameter-to-period ratio is increased because volume of the material is reduced and phonon scattering surface is increased. However, the samples with aligned and staggered lattices have the same volume and the same surface area. For this reason, at the microscale...”

B) p.8: While it is right that reaching tangentially a surface allows for specular reflection, one should keep in mind that the probability of such a large angle is usually very small.

In general, indeed, it may seem that only a small portion of phonons would hit the side wall of holes tangentially. However, it is so only for maybe first few rows of holes. Supplementary Figure 12b shows that as the number of hole rows increases, phonons develop the directionality and after ten rows of holes, most phonons travel parallel to heat flux direction ($\theta = 0^\circ$). Thus, if they meet a hole surface, it happens quite tangentially. The same idea is illustrated in Tang *et al.* [JAP 114 184302 (2013)] in Fig. 8, where angle between the hole surfaces and local phonon fluxes is less than 5° (i.e. $\alpha \geq 95^\circ$). To clarify this idea, we added on page 9: “Such tangential incidences often occur^{41,42} in the passages between the holes of the aligned lattice as phonons develop directionality passing through the structure (Supplementary Note 6).”

C) Fig. 4d: What could explain that the required roughness increases when the length increases? In principle the roughness does not change with length.

We also do not suppose that roughness changes with the length of the nanowires as all the structures were fabricated simultaneously. By this plot, we only aim to show that the experimental length dependence shows the same trend as would be expected theoretically for the roughness in the 2 – 3 nm range, so it is in agreement with our estimations of roughness. In this revision, we have improved the simulations by considering real phonon angular distribution (see R2.13), and now the agreement is better. However, we should note that precision in measured value of Δ is certainly not enough to attribute some meaning to the 2 – 3% disagreement with the simulations and discuss possible physical explanations of that. That is why we only briefly acknowledge that “the trends are qualitatively consistent with our experimental data”.

D) Fig. 4e: Why there is no simulation for the temperature dependence? Was there something strange there in the simulated results there?

Our simulations are strictly valid only at low temperatures (≤ 20 K), because of the Debye model used. Thus, we did not try to perform temperature-dependent simulations. This interesting question is similar to the question 2 of reviewer #1. Please refer to that question for more details on temperature-dependent simulations.

E) Line 159: mentioning 50 nm at room temperature from the manuscript by Henry and Chen is too large as the contributions of these wavelengths seem very small (less than 5%).

As you suggested, we have contacted Prof. Henry and he advised to use Esfarjani’s data as more rigorous. We also found another reference that seems to confirm the same phonon wavelength range [Malhotra and Maldovan, Sci. Rep. 6, 25818 (2016)]. Thus, we keep only the 0.5 – 5 nm range.

F) Fig. 5(e-g): The definition of quantity of the ordinate axes is not given in the core of the manuscript. Please indicate it.

Indeed, this was missing, we added a sentence: *“To compare the experimental data with predictions of Monte Carlo simulations (Methods), in Fig. 5e–g we plot the decay times normalized by their average value.”*

G) Line 244: Please mention the criterion used for determining the size of the hot spot in the text also: the full width at half maximum, if I understood correctly.

Yes, it is correct. We corrected as: *“focal point, located 0.5 μm away from the structure, forming a hot-spot with full width at half maximum of 115 nm”*

H) Line 187: replace “it does” by “it should” (no evidence, only a supposition)

We corrected

I) Line 278: “could” instead of “can”: only supposition.

We corrected

J) Line 98: typo, remove “the” in “the most data”.

We corrected

K) Typos everywhere (at least 5x in the manuscript and in the Supplementary Material): Planck, with a “c”. Example: Lines 351

Thank you for pointing out the typos. We corrected all of them in the revised manuscript. We also used English correction service to improve this version of the manuscript.

L) References: Please harmonize, either writing all the author names or first author “et al.” The same comment applies for the Supplementary Material.

This reference style is provided by the journal guidelines: “All authors should be included in reference lists unless there are six or more, in which case only the first author should be given, followed by 'et al.'” Therefore, we keep as it is, but we thank you for this kind suggestion.

The following remaining weaknesses have been found in the current version of the Supplementary Material

S.A) Monte Carlo simulations. When a surface emits phonons isotropically, as is the case for a blackbody in radiation, the random angle theta is usually drawn according to a cosine distribution for phonon bundles. Here it is apparently drawn without this cosine distribution, so that more phonon bundles are launched with large angles than it should. This is a strong issue, as it fastens the heat dissipation from the aluminium surface. The authors should correct their simulations or prove that this has no impact on the results. This is a key point.

Thank you for this important comment. Although, we are aware of cosine distribution of a blackbody radiation and we already use cosine distribution for the diffuse scattering events on geometrical boundaries, we used an isotropic emission because we were unsure if this is applicable for aluminium/silicon boundary. As suggested by reviewer, we implemented the cosine distribution for the emitted phonons following the Lambert’s cosine law [Klitsner et al, PRB 38, 7576 (1988)]. The results are shown in the figure below for a membrane (without holes) and a converging lens.

No difference in the energy curves is observed between the isotropic and the cosine distribution for the two situations. This is certainly because the change in the distribution only impacts only the first boundary scattering events. The curves are shown in absolute values to show that the superposition is not due to the normalization process that we usually apply for the analysis. Finally, although the simulations are certainly improved by the new emission process suggested by reviewer, it has no visible impact on the results. The Lambert's cosine law emission process has been added to the Supplementary Information.

S.B) The comment on the fact that it is not possible to reproduce numerically the exact decay time is very intriguing. I do not at all understand why the decay should not be the same depending on the power of the incident light if the heat equation in the linear regime is considered. This is a key point that may shake the whole demonstrations in the manuscript.

The fact that the simulated decay times are shorter than experimental ones is not surprising since our simulations are not capturing the physics of heat transport in its entirety. For example, as we use the Debye approximation of linear dispersion relation, we do not capture the flattening of the bands where the group velocity is low but the density of states is high.

First, let us clarify the misunderstanding. More power implies an temperature increase that shifts the phonon spectrum toward higher frequencies and thus increases the probability of internal scattering and diffuse scattering on the boundaries. With more scattering events, the phonon trajectories become chaotic and the decay time becomes longer. This effect is non-linear as the phonon-phonon scattering rate expressions are T^4 and T^3 dependent. An illustration is given in the question 2 of reviewer #1 in which we show that as the wavelength decreases the decay time increases. However, we acknowledge that this effect is not enough to explain the difference between experiments and simulations.

We carried out further investigations and established that the ratio of the experimental decay time over the simulated decay time is always a constant ~ 9 , as shown in the figure below. It means that we only have a scaling factor between the experiments and the simulations. Furthermore, if we look at the right vertical axis on the panel b Question 2 of reviewer #1, we see that there is a factor ~ 3.5 between the decay time for $\lambda = 25$ nm and the decay time for $\lambda = 2$ nm. This is coherent with the factor ~ 3 observed experimentally between the decay times at 4 K and the decay times at 300 K.

From these two facts, we conclude that although the simulations miss some background phenomena that affect evenly all the structures, the physical changes observed experimentally are well reproduced by the simulations. When we divide by the average decay time in Figure 5 in the main text, the background is canceled so that experiments and simulations fall into agreement.

Finally, let us emphasise that this is mostly experimental work, and the simulations mostly serve to illustrate the phonon trajectories and thus explain the experimentally observed differences between samples. Even if the simulations lack some phenomena that take place in reality, the experimental data remain quite clear and we do not see any alternative explanation for the experimental data.

S.C) Fig. S5: Please highlight that the ordinate axis is in logarithmic scale, so that the data dispersion is not easily observable. There are variations by factors 2 or 3 for some values of the neck...

We have corrected as: *“Although the plot is logarithmic, most of the data points seem to form a common trend...”*

S.D) Fig. S11: the captions of the abscissa are cut.

This was an artefact of pdf conversion, we will make sure that it is correct in the final submission.

S.E) Line 27: “non-Fourier” or “non-diffusive”, but not “non-classical” as nothing is quantum here.

We corrected as non-diffusive.

S.F) Line 30: Typo, remove “do”

We corrected into “no”.

S.G) Line 79: remove “most”

We removed.

S.H) Lines 94-95: replace the erroneous sentence by “thus using Eq. S1 to estimate the specular parameter seems reasonable.”

We corrected

I now review the answers in the rebuttal letter.

Reviewer #1

R1.0. “Questions about thermal phonon coherence: why not accounted?”

The authors explain that there that staggered structures stay periodic and indicate that they found similar dispersion curves. It would be good to also provide these data to the reviewers so that this claim can be analyzed. In particular, the authors do not indicate up to which frequencies they computed the data.

Thank you for your interest to our other theoretical works. To illustrate our point, let us plot average group velocity for aligned and staggered lattices ($a = 160$ nm, $h = 145$ nm, $r/a = 0.3$) together with that of unpatterned membrane calculated from the obtained band diagrams.

We can see that whereas there is some difference between the lattices at low frequencies, at higher frequencies, they are quite similar and both lattices strongly suppress heat conduction as compared to unpatterned membrane. Thus, the “phononic effect” is present in both lattices. Due to computational limitations, we calculated phonon dispersion only up to ~ 200 GHz, which is enough to describe heat conduction at 1 K, but we see no reason why at 4 K this situation would be dramatically different. Recently, we have investigated the wave effects that might occur in different lattice types [Anufriev and Nomura PRB 93 045410 (2016)] and found that even lattices as different as hexagonal, square and honeycomb have very similar impacts on thermal conductance, and that the other geometrical parameters are more important than the lattice. Thus, it is not very surprising that aligned and staggered lattices have very similar impact on band structure.

R1.1. “Distribution of phonons not accounted”

The authors argue that the actual distribution is not known. The answer is not really satisfactory. Planck distribution is cut by the Debye wavelength, there is no need to consider lower wavelengths. This should actually provide already interesting results. It would be interesting to see if the results are maintained by considering the not-too-large wavelength, knowing that possible large wavelengths would make the effects stronger.

We conducted additional analysis and found that longer wavelength indeed yields a stronger focusing effect. For the details, please refer the question 2 of reviewer #1.

R1.2. “Roughness vs TDTR?”: the question and the answer are not really useful.

R1.3. “If mention of rectification, then quantify it”

The authors have wisely erased the speculative mention of rectification, as it is not the message that is conveyed here.

R1.4. “Specularity and correlation”

A new section has been added, which is good. One could however argue that RIE induces correlation as was shown by SEM images in some papers in the past. But maybe the associated wavelength is too large.

R1.5. “Clarify roughness values”: this has been done.

R1.6. “Errors in figure numbers”: this has been corrected.

R1.7: “Directionality and “directional fluxes””

Contrarily to this reviewer, I was not shocked by the previous way it was written. The new one is also reasonable.

Reviewer #2

R2.1. “Wave character is not considered”: the answer is satisfactory.

R2.2. “Large delta T in comparison to temperature at 4 K.”

The authors indicate that despite this fact nothing special seems to appear in comparison to higher temperatures. Another comment is given, stating that the decay times depend also on the geometry, not only on the thermal conductivity, and that inverting the geometry is sufficient to change the decay time even through staying in the linear regime. Such comment should be inserted at least in the Supplementary Information. I suppose that it has been observed in linear FEM simulations.

Here, we should emphasize that while thermal conductivity of a certain region can be extracted from decay time via FEM modelling of the same geometry, the decay time itself depends on the geometry of the sample. The decay time is a measure of heat dissipation from the central region. Thus, it is not surprising that converging lens, for example, shows slightly different decay time than that of the diverging lens. Indeed, in the converging lens heat has only five 50-nm-wide channels to pass from central region to the lens, whereas in the diverging lens, there are much more channels to enter the lens, but only five to leave the lens. Thus, the dynamics is different and decay times are not necessarily the same.

This difference also appears in FEM simulations. Alternatively, for example, if we move the lens structure (i.e. all holes) closer to the metal pad, the decay time also naturally changes even in FEM simulations.

For these reasons, it is so important to compare only very similar structures, thus the entire study is designed as comparative, and we discuss only changes in decay time, but not the absolute values.

We have added a comment to Supplementary Information: *“The measured decay times are strongly dependent on the geometry of the structure and placement of the holes;”*

R2.3. “Simulations with the nanowires are not very well matching the experimental data”: the authors state that 3D Monte Carlo now better matches.

R2.4. “How averaged are the data? Dispersion?”

How the data are averaged has been clarified in the rebuttal letter. It would be good to clarify that even better in the manuscript.

Indeed, it is a good idea, we have added it in the Methods: *“To eliminate inaccuracies that might be caused by the laser beam alignment, each sample was measured twice on different days, whereas converging lens and reference samples were measured three and four times, respectively. Only some of the 160-nm-period samples could not have been successfully measured more than once due to fragility of the structure”*

R2.5. “Error in labelling”: this has been corrected.

R2.6. “How is the roughness measured?”

The answer is OK. Data on the tip radius of curvature should be given at least in the Suppl. Mat..

We have added this information in the “Roughness consideration” part of the Methods section.

R2.7. “Why not simulating the lenses with FEM?”

The authors indicate that they would need to include thermal conductivity that locally depends on size/thickness/width of the structure and that it is difficult. They estimate that the Monte Carlo simulations are sufficient to find the trend. While I agree that the Monte Carlo simulations seem to show the same trend, the comment on the fact that even in the diffusive regime the decay time depends on the orientation leads to require these FEM simulations. Maybe they cannot be quantitative, but it would be important to see if they provide another type of trend or if the trend is the same than that of the Monte Carlo data. Indeed, if they have the same trend, then nothing is demonstrated...!

Let us consider the FEM model in the figure below. To at least partly account for the different regions of thermal conductivity we set 3.5 times lower thermal conductivity in the blue region, according to [Anufriev *et al.* PRB 93 045411 (2016)], and twice lower thermal conductivity in the wires and the slit. The simulation results, shown by blue curves, seem to be very similar for converging lens, diverging lens and even reference membranes. This resembles neither our experimental results nor Monte Carlo simulations. If in the reference structures the simulated trends somewhat pass through the error bars, in case of converging lens, experimental and MC trends are clearly steeper. Moreover, whereas FEM trends are nearly the same for all three types of samples, the experimental trends are clearly different, thus we conclude that the experimental results cannot be explained in terms of purely diffusive transport.

- R2.8. “Simulations in the stationary regime and not in the transient one”: the authors now perform transient simulations, so it seems now more reasonable.
 Another comment on the transmission as a function of frequency is not really relevant but is not key to the work.
- R2.9. “2D vs 3D Monte Carlo simulations”: the authors have modified their simulations and now perform 3D simulations.
- R2.10. “Roughness on membrane surfaces”: it is now taken into account.
- R2.11. “Main contributing wavelength of 3 nm seems too large”: this has been modified.
- R2.12.” Structure is too small so that it is a filter: there is no diffusive regime.”
- The authors know that the mean free path at 4 K is much larger than their structure size (in fact their expression for the relaxation time should help them to evaluate it!). It would be good to determine the effective mean free path in their structures by accounting diffuse scattering at

boundaries. However, I verified and there is no determination of thermal conductivity at 4 K, thus there is no problem in the manuscript.

R2.13. "Why taking a Lorentzian curve and not the actual data?"

The authors considered that Fig. 3d can be fitted by a Lorentzian curve. One can see that the bottom part will be certainly underestimated in this case, so the effect of the directionality might be artificially increased with this procedure...

The same question than S.A. asked higher about the cosine direction for the "emission" of phonons was not answered.

It is true that by simulating only the peak region we overestimate the directionality effect. To improve our results, we implemented a Von Neumann rejection algorithm (aka Acceptance-Rejection method) in order to simulate directly the angles distributions for θ and φ shown on Figure 3 of the main text and Supplementary Figure 10. The results are shown below.

As expected, due to the presence of phonons with large angles, the direct simulation of the angle distributions leads to a lower $\Delta_{Uncoupled}^{Coupled}$ as compared to our previous simulations with the Lorentzian of the peak. The difference between the two methods is $\sim 2\%$ for the nanowire length of 1 μm and it reduces as the nanowire length increases. The convergence of the direct and Lorentzian methods is due to the increase of the number of scattering events with the length that progressively reduces the impact of the initial angle distribution.

The direct simulations of the angle distributions have now replaced the previous simulations in Figure 4d of the main text.

R2.14. "Normal vs Umklapp scattering in the simulations"

Only Normal scattering is considered because all the simulations were performed at 4 K. Avoiding simulations at 300 K is a weakness of the manuscript.

For the expanded comments on the matter, please refer to the question 2 or reviewer #1. In summary, the main results of our paper are at 4 and 30 K thus, we created our Monte Carlo code for low temperatures. We would like to extend the code to high temperature in a future work.

R2.15. "Local effective temperature"

I think that providing the local effective temperature (what will be useful to compare with what is known) is important. Of course the authors should explain that this is not a thermodynamic equilibrium temperature (clear from the non-isotropic temperature distribution!), but that this is just the local energy. By the way, the whole paper is based on

anisotropic distribution function for phonons (so angular distribution of energy is plot), but the distribution in energy (maybe along some direction) is not shown.

Note that the local energy is not proportional to T^4 for phonons at room temperature.

Following the previous comment on the temperature, we looked more closely to this point. The T^4 proportionality with the energy is intimately linked with the Planck distribution for phonons. We tried to verify if the energy distribution in one cell (defining one pixel of the map) at the focus point follow a Planck distribution. The results are not conclusive, as you can see on the figure below, because there is not enough phonons to have a good spectrum. Therefore, we think that drawing any conclusion based on the temperature would be very unreliable. Furthermore, even if we blindly apply the formula $E \propto T^4$, as the calculated temperature would not correspond to any practical physics, we don't think it would be comparable to what is known.

First, we would like to mention that the computational requirements to make a full study of this kind are almost impossible to meet with our equipment. Indeed, the obvious solution would be to increase the statistics to obtain a conclusive energy distribution. For information, one simulation takes around one week. We estimate that this time should be multiplied by 10 to obtain satisfactory statistics and should be again multiplied as we should extract the energy distribution of hundreds of cells to make a map with a much lower resolution than we have now (there are $150 \times 200 = 30,000$ cells currently).

Second, even with smooth energy distributions, there is almost no chance that we can easily deduce the temperatures of the cells. Theoretically, the Planck distribution should shift toward the low frequencies as the temperature decreases from the heated part of the structure to the cold reservoir. However, in the previous response letter, we showed the energy transmission is overall constant for all frequencies. If we assume that this global behavior is also valid at the local scale, for a cell of 5×5 nm, then it means that the local distribution is proportional to the initial distribution (the factor of proportionality would depend on the location of the cell in the nanostructure). It means that most probably we would not observe any significant frequency shift. However, a lower energy necessarily means a lower temperature but the question would be how to link energy and temperature. As for us, all these issues are not so related to our paper and have no impact on our conclusions. We believe that the direct output of the energy as given by the Monte Carlo simulations is the most reliable and correct data and therefore should be used in the manuscript.

As suggested by reviewer, we have plotted energy profiles along other directions. The blue and green lines of the energy profiles on the figure below are obtained for the dashed blue line and the dashed green line, respectively, as represented on the energy map. We only map the nanostructured region but we can imagine that the energy level at the heated region must be close to the one we see for the smallest x . Therefore, the intensity of the hotspot is ~ 10 times lower than the heated region. The increase of energy observed for $x \sim 3.2 \mu\text{m}$ and $x \sim 4.0 \mu\text{m}$ correspond to the regions just before the

passage between two holes where many phonons are backscattered on the boundaries of the holes. This effect decreases as a function of the number of holes the phonons cross as they acquire the directionality of the direct passage between the holes.

R2.16. "Specularity factor as a function of temperature": this has been done.

R2.17. "Pictures of the devices": they have been added.

R2.18. "Heat transport is ballistic at micrometers": the new wording is reasonable. Note that the reference is not complete ("PhD thesis" missing).

We replaced the reference to the thesis by the reference to Maire *et al.* [Sci. Rep. 7, 41794 (2017)], which has been recently published.

R2.19. "Strange to have an axis below 0": Still the case. I leave this open to the authors.

R2.20. "Cross-plane vs in-plane studied in a reference": Yes, the authors are right.

R2.21. "Last section to be softened": OK.

R2.22. "Filling factor in the phononic crystals"

A new section has been added: it is clear now. However in the manuscript it could be good to always show that the effects are stronger than those expected from simple effective medium theory.

R2.23. "Heat conduction" and not "heat transport": this has been done.

Typos underlined are also corrected.

Reviewer #3

R3.1. "TDTR provides more information than what is used. Simulations difficult to compare with experiments. Analytical solution for heat diffusion possible?"

The authors do not directly answer to this not-so-clear question. For instance, the authors do not reproduce exactly numerically the experiments: the intensity is always normalized, which weakens the trust of the reader in the numerical reproducing of the data. Some improvement or comment is in order here in the manuscript.

The normalization of the signal is quite common in TDTR measurements because, at least in our experiments, the decay time constant does not depend on the intensity of the signal, but intensity of the signal can change from one sample to another due to difference in laser alignment. Thus, the data is normalized simply for convenient comparison. However, we could as well fit original curves. In case of FEM simulations, the decay constant also does not depend on the intensity as there is no non-linear mechanisms at play. The only real difference between experiment and simulation is that experimentally we measure changes in laser intensity induced by changes in reflectance, which are in turn induced by changes in temperature, whereas in simulation we directly measure changes in temperature. However, they are directly proportional via thermoreflectance coefficient, thus there is nothing to weaken reader's trust.

We explain this in the main text: *“the pulsed pump beam periodically heats the aluminium pad, while the change in its reflectance ($\Delta R/R$), caused by the heating, is monitored by the continuous-wave probe beam. Since the change in the reflectance is proportional to the change in the temperature via the thermoreflectance coefficient, we can record relative changes in the temperature ($\Delta T/T$) of the aluminium pad in time (t).”*

And also mentioned in Supplementary information: *“Using the thermal conductivity of the phononic crystal region as a free sweeping parameter, we simulate heat dissipation through the structure and monitor changes of temperature in the metal pad.”*

R3.2.” Decay always exponential?”

The authors indicate that the decay is always exponential if the regime is diffusive, but do not tackle other regimes. This points to R2.12: if there is no diffusive regime, why should the decay be exponential? An answer can be that the decay will always be diffusive finally at some location, maybe far from the structure. Another answer is also that all observed decays seem exponential...

The exponential decay is probably only an approximation. In case of phononic crystals (i.e. with holes), the overall transport is still mostly diffusive, and ballistic only in one direction between the holes, so the exponent is a good approximation. Interestingly, in unpatented membranes, in which ballisticity should be strong, we see a deviation from exponential decay. In the figure below, the first 0.75 μs after the end of the pump pulse (1 μs) the decay is faster than can be fitted with the same exponent as rest of the curve.

Thus, we corrected Methods to precise that this is an approximation: *“This function can be approximated well by an exponential decay”*

R3.3. “Thermal energy distribution”

The authors explain the procedure from the Monte-Carlo simulation, but it would be good to provide expressions as a function of the distribution function. This is easy. Writing this part will also make easier the answer to R2.15.

Thank you for the suggestion. The mathematical expressions of the distribution functions are now given in Supplementary Note 4.

R3.4." Broad distribution of phonon wavelengths": the answer of the authors is satisfactory.

R3.5." Decay time considered includes both phononic crystal and nanowires": the authors have studied this effect experimentally and numerically. Since the goal is to see the coupling, I feel that what has been done is sufficient.

R3.6."Not understanding the wire simulation": this is well explained.

R3.7. "More than half of the phonons have MFP larger than 1 micron": this is clear.

R3.8. "Phonon model is basic".

This is a good point from the referee. The authors indicate that they perform the simulations at low temperature to avoid issues at room temperature. While not having the simulations at room temperature is a weakness, there is some logic in the manuscript.

R3.9. "Values of the thermal conductivities considered": this is well explained. If I understood correctly, these values are however not used at low temperature.

R3.10. "Geometrical factor is maybe geometry-dependent...": the text in the Supplementary Material is now useful.

R3.11. "Steady-state vs transient state"

The authors indicate that a previous explanation was incorrect, but that it is corrected in the current version.

To conclude, the main remaining points that need to be tackled are S.A, S.B, R2.2, R2.7, R2.15 (R3.3), R3.1, R3.2. The reviewer would also have preferred to see some data for R1.0. Once all these points have been addressed, the manuscript could be published.

We highly appreciate your suggestive advice to this work and taking a lot of your time to improve our manuscript. We addressed all the points and hope that you will find the revised manuscript to be suitable for publication.

Reviewer #3:

I thank the authors for their careful consideration of my comments and those of the other reviewers. They have made their already strong manuscript even better. I recommend publication in Nature Communications.

Comments as related to my original numbering scheme

2. The temperature response provided by the authors in the rebuttal, as they themselves note, is based on a solution of the thermal diffusion equation. As such, it is inherently diffusive. The central focus of this work is on ballistic transport. As such, it seems that the exponential decay is an approximation, although apparently a good one.

Indeed, the exponential decay is probably an approximation. In case of phononic membranes (i.e. with holes), although ballistic fluxes exist between the holes, the overall heat transport is still mostly diffusive, so the exponent is a good approximation. Interestingly, in unpatented membranes, in which ballisticity should be strong, we see a deviation from exponential decay. In the figure below, the first 0.75 μs after the end of the pump pulse (1 μs) the decay is faster than can be fitted with the same exponent as the rest of the curve.

Thus, we corrected the Methods to precise that this is an approximation: “*This function can be approximated well by an exponential decay*”

4. The range of wavelengths reported in Ref. 40 (up to 50 nm) does not seem correct. Could it be due to the use of an empirical potential and not first principles? I suggest that the authors contact Asegun Henry (now at Georgia Tech) to ask his opinion.

Thank you for this suggestion. We contacted Prof. Henry and he advised to use Esfarjani’s data as more rigorous. We also found another reference that seems to confirm the same phonon wavelength range [Malhotra and Maldovan, *Sci. Rep.* 6, 25818 (2016)]. Thus, we keep only the 0.5 – 5 nm range.

5. I appreciate the extra analysis. While implicit, I think that it would be worth mentioning in the text that the tau for these structures includes the effect of the phononic crystal and the nanowires.

We have added a mention of this fact: “*These fluxes enforce ballisticity in short nanowires and thus cause the difference between coupled and uncoupled samples, though the effect is partly masked by the phononic crystal part before the nanowires*”

10. Thank you for providing this additional information in the SI. I note that Jain et al. (Ref. 27 of the main text) found that the Hashin factor, $(1-\phi)/(1+\phi)$, provided a good comparison to their COMSOL simulations of similar structures as considered here.

Yes, indeed in the literature one can find this and other factor too. We tested them, and found that the form with $1/2$ make the best fit for our data. However, the other factors are also quite close, and particularly the Hashin factor gives rather good agreement too. We have added this to the discussion in Supplemental Information: “*Note that the factor in the form of $F(\phi) = (1 - \phi) / (1 + \phi)$ can also be found in the literature and also gives a satisfactory agreement.*”

New Comments

i. Page 6, line 118. Better to say “effective thermal conductivity” due to the presence of ballistic effects.

We replaced the term with effective thermal conductivity and also changed corresponding values.

ii. Page 8, line 155. When the authors write “diffuse scattering in random directions,” what distribution is being sampled? It should be a Lambert cosine.

Yes, the directions of the phonons diffusively scattered are sampled using the Lambert's cosine law. We precised this fact in the Supplementary Information.

iii. Page 9, line 173. I suggest changing “strengthens” to “increases” to avoid ambiguity.

We agree and changed the word.

iv. Page 12, line 239. Word “however” is not appropriate here.

The word has been omitted.

We highly appreciate your suggestive advice to this work and recommendation for publication.

List of main changes

1. We sent the manuscript to a language editing service, which helped us to improve quality of writing and correct remaining typos and errors.
2. In multiple places, we changed the wording and added explanations in response to reviewers' suggestions.
3. We conducted complimentary Monte Carlo simulations in order to make figures smoother and increase resolution of thermal maps.
4. We improved the simulations on nanowire-coupled samples by considering real phonon angle distributions instead of their Lorentzian approximations. The agreement with the experiment is now better.
5. The Supplementary information has been extended to include the data requested by reviewers.

Reviewers' comments:

Reviewer #2 (Remarks to the Author):

I first review the new answer by the authors.

Referee #1

Point 1. "Low-frequency phonons reflected irrespective of temperature".
The answer is satisfactory.

Point 2.1. "Phonon spectrum should be considered"
No answer.

Point 2.2. "Inconsistency between the temperatures of the experiments and that of the simulations"

The new version shows results at various wavelengths. While I don't think that modifying the code would have been an issue and I regret that temperature-dependent results are not provided, the new version of the Suppl. Mat. is acceptable.

Note that the temperature of the experiment performed in Fig. 4 (4 K following this letter) is apparently not given in the main text and should be provided clearly. If the authors have verified that 30 K does not change anything in Fig. 3 in comparison to 4 K, they should add on line 141 the following sentence: "We stress that performing these simulations at 30 K would not change the main observations that follow".

Referee #2

Most of points A-L have been addressed.
Point C stays unclear but can left as is.
Point SA has been addressed seriously.

Point SB. "Decay times in the experiment and the simulations are very different".
This stays an important problem. The flattening of the bands mentioned by the authors cannot explain a factor 9 difference. In panel b of the answer to question 2 of Reviewer #1 a factor 3-4 only is found and this can neither explain the difference as acknowledged by the authors. In addition, there is no convincing argument that would justify that the possible "background" can be added or removed as explained by the authors.

As a result, I asked the authors to change Fig. 5(e-g) by the curves to provide the experimental data without normalization. They can add additional curves or insets with the simulations also without normalization to show that the trends are similar but the reader should see that the decay times found are VERY different. The current sentence in the text is not sufficient at all. Finally, the Figure in the rebuttal letter should also be added in the Suppl. Mat..

Points SC-SH have been addressed.

Point R1.0. "Coherence not mentioned"
The authors provide a curve at low frequency. It should also be added in the Suppl. Mat. and referred to in the main text.

Point R1.1. OK.

Point R2.2. "Temperature does not impact much, while geometry impacts strongly"
Minimal comment has been added.

Points R2.4 and R2.6 have been addressed.

Point R2.7. "Simulate the lenses with FEM"

The authors have done so and the results look convincing. The results not normalized should be included in the Suppl. Mat..

Point R2.13 has been addressed nicely.

Point R2.14 is the same as Point 2 of Ref. #1

Point R.15. "Local effective temperature"

The authors provide a spectrum, which is discretized and shows that a satisfying statistics is not achieved spectrally. A solution could have been to smoothen the data (telling the reader).

While I would prefer them to provide the value of the "effective temperature", I can accept them to provide only the local energy in Joule as they do in their second Figure. The color bars of Figs. 3(a-c) and 5(b-d) should provide these data.

In addition, the additional profiles along some directions are extremely useful as they provide an idea of the strength of the hot spot. I suggest it to be added into the paper.

Points R2.18, R3.1 and R3.2 have been addressed in a satisfactory manner.

Points R3.3. "Thermal energy distribution"

The section of the Suppl. Mat. on the Monte Carlo procedure have been expanded. Some of the answers of R2.15 could be included.

I also noted few issues in the current version of the manuscript.

N1. I noted that the values on lines 122-123 are strongly modified. Could these differences be explained?

N2. I have difficulties in trusting the sentence on lines 186-187 in the Suppl. Mat. (I suppose they refer to Suppl. Fig. 11). Previously it was said that there is no temperature effect (Point R2.2). So what is right?

Typos

N3. Line 84. The meaning has been changed. Please revert to the original version "in which every second row of holes is shifted".

N4. I don't think that "s.d." is defined in the text.

N5. Line 101: "However" to be replace by "Here"

N6. Line 146: Remove "the" before "exit".

N7. Line 286: Probably what is meant is "below 20 K, temperature-dependent simulations should become the subject of future work".

N8. Line 179 of Suppl. Mat.: "is rapidly decreased"

As a conclusion, the authors need to perform the changes related to Points 2.2, SB, R2.15 and few other additions to the Suppl. Mat. prior to publication. I also expect answers to Points N1 and N2.

Point 2.2. “Inconsistency between the temperatures of the experiments and that of the simulations”

The new version shows results at various wavelengths. While I don't think that modifying the code would have been an issue and I regret that temperature-dependent results are not provided, the new version of the Suppl. Mat. is acceptable.

Note that the temperature of the experiment performed in Fig. 4 (4 K following this letter) is apparently not given in the main text and should be provided clearly. If the authors have verified that 30 K does not change anything in Fig. 3 in comparison to 4 K, they should add on line 141 the following sentence: “We stress that performing these simulations at 30 K would not change the main observations that follow”.

We also regret that we cannot conduct this analysis at the moment. However, we have learned that another group decided to conduct such a study to explain our experimental results, thus we hope that their results will appear in the literature within a year.

The temperature of the experiment in Figure 4 is stated in the line 223: “*as a function of the nanowire length (L_{NW}) at 4 K*”. As for the Figure 3, we agree with the proposed modification and added this sentence on line 141.

Point SB. “Decay times in the experiment and the simulations are very different”

This stays an important problem. The flattening of the bands mentioned by the authors cannot explain a factor 9 difference. In panel b of the answer to question 2 of Reviewer #1 a factor 3-4 only is found and this can neither explain the difference as acknowledged by the authors. In addition, there is no convincing argument that would justify that the possible “background” can be added or removed as explained by the authors.

As a result, I asked the authors to change Fig. 5(e-g) by the curves to provide the experimental data without normalization. They can add additional curves or insets with the simulations also without normalization to show that the trends are similar but the reader should see that the decay times found are VERY different. The current sentence in the text is not sufficient at all. Finally, the Figure in the rebuttal letter should also be added in the Suppl. Mat..

We followed the reviewer's suggestion and changed Fig. 5(e-g) so that now it shows absolute experimental values on the left axis and absolute simulated decay times on the right axis, without normalization. We keep the same range of $\pm 13\%$ of the average value to show that the trends are similar.

To emphasize the difference between the theory and the experiment in the main text, we added on line 275 of the main text: “*Although in absolute values the simulated decay times are much shorter than the experimental ones (Supplementary Note 5), the trends are generally in good agreement*”. Moreover, we explained this disagreement in Supplementary Note 5 and added the figure from the rebuttal letter.

Point R1.0. “Coherence not mentioned”

The authors provide a curve at low frequency. It should also be added in the Suppl. Mat. and referred to in the main text.

We agree and added this figure in Supplementary Note 7. We referred to it in the main text as well.

Point R2.7. “Simulate the lenses with FEM”

The authors have done so and the results look convincing. The results not normalized should be included in the Suppl. Mat..

We agree and placed the figure and corresponding discussion in Supplementary Note 8 and referred to it in the main text.

Point R.15. “Local effective temperature”

The authors provide a spectrum, which is discretized and shows that a satisfying statistics is not achieved spectrally. A solution could have been to smoothen the data (telling the reader).

While I would prefer them to provide the value of the “effective temperature”, I can accept them to provide only the local energy in Joule as they do in their second Figure. The color bars of Figs. 3(a-c) and 5(b-d) should provide these data.

In addition, the additional profiles along some directions are extremely useful as they provide an idea of the strength of the hot spot. I suggest it to be added into the paper.

To reach the agreement with the reviewer in this long lasting question, we replaced the arbitrary units by values in Joules in figures with energy maps. Also, as suggested, the figure with the profiles has been added as Supplementary Figure 16.

Points R3.3. “Thermal energy distribution”

The section of the Suppl. Mat. on the Monte Carlo procedure have been expanded. Some of the answers of R2.15 could be included.

We agree and added the figure with the profiles, part of R2.15, as Supplementary Figure 16 along with the associated discussion.

I also noted few issues in the current version of the manuscript.

N1. I noted that the values on lines 122-123 are strongly modified. Could these differences be explained?

Reviewer #3 asked us to give values of effective thermal conductivity instead of regular thermal conductivity, thus the values are different from the previous version. However, this does not affect the discussion as the relative difference remains the same. For more details about the difference between effective and regular thermal conductivity, please see Supplementary Note 1.

N2. I have difficulties in trusting the sentence on lines 186-187 in the Suppl. Mat. (I suppose they refer to Suppl. Fig. 11). Previously it was said that there is no temperature effect (Point R2.2). So what is right?

Thank you for pointing out this issue, and sorry for the inaccuracy, we indeed referred to Fig. 11 (not 9). As we concluded, we do not fully understand the reasons behind different decay times in the simulation and the experiment, thus we decided to omit these confusing sentences on lines 186-187, and instead add a discussion on the difference between experimental and simulated decay times together with the figure showing constant $\tau_{\text{exp}}/\tau_{\text{MC}}$ ratio for all structures.

Typos

N3. Line 84. The meaning has been changed. Please revert to the original version “in which every second row of holes is shifted”.

Thank you for noticing, we corrected.

N4. I don't think that "s.d." is defined in the text.

This abbreviation of standard deviation is suggested by journal guidelines. Thank you for noticing, we will check with copy editors.

N5. Line 101: "However" to be replace by "Here"

We agree and changed.

N6. Line 146: Remove "the" before "exit".

We agree and removed.

N7. Line 286: Probably what is meant is "below 20 K, temperature-dependent simulations should become the subject of future work".

We agree and corrected.

N8. Line 179 of Suppl. Mat.: "is rapidly decreased"

Thank you, we corrected.

As a conclusion, the authors need to perform the changes related to Points 2.2, SB, R2.15 and few other additions to the Suppl. Mat. prior to publication. I also expect answers to Points N1 and N2.

We addressed all the requested points and hope that the revised manuscript is finally ready for publication. Finally, we would like to express our gratitude to all reviewers again for their help, comments and countless valuable suggestions that certainly improved our manuscript a lot. We highly appreciate all the time and efforts put into reviewing our work during this long but productive time. Thank you.

Sincerely,

Roman Anufriev